# Pre-configuring chromatin architecture with histone modifications guides hematopoietic stem cell formation in mouse embryos

Chen C. Li [1,5], Guangyu Zhang[2,5], Junjie Du[2], Di Liu[1], Zongcheng Li [3], Yanli Ni[3], Jie Zhou[3], Yunqiao Li[2], Siyuan Hou[4], Xiaona Zheng[2], Yu Lan [4✉], Bing Liu [2,3,4✉] & Aibin He [1✉]

The gene activity underlying cell differentiation is regulated by a diverse set of transcription factors (TFs), histone modifications, chromatin structures and more. Although definitive hematopoietic stem cells (HSCs) are known to emerge via endothelial-to-hematopoietic transition (EHT), how the multi-layered epigenome is sequentially unfolded in a small portion of endothelial cells (ECs) transitioning into the hematopoietic fate remains elusive. With optimized low-input itChIP-seq and Hi-C assays, we performed multi-omics dissection of the HSC ontogeny trajectory across early arterial ECs (eAECs), hemogenic endothelial cells (HECs), pre-HSCs and long-term HSCs (LT-HSCs) in mouse embryos. Interestingly, HSC regulatory regions are already pre-configurated with active histone modifications as early as eAECs, preceding chromatin looping dynamics within topologically associating domains. Chromatin looping structures between enhancers and promoters only become gradually strengthened over time. Notably, RUNX1, a master TF for hematopoiesis, enriched at half of these loops is observed early from eAECs through pre-HSCs but its enrichment further increases in HSCs. RUNX1 and co-TFs together constitute a central, progressively intensified enhancer-promoter interactions. Thus, our study provides a framework to decipher how temporal epigenomic configurations fulfill cell lineage specification during development.

[1] Beijing Key Laboratory of Cardiometabolic Molecular Medicine, Institute of Molecular Medicine, Peking-Tsinghua Center for Life Sciences, Peking University, 100871 Beijing, China. [2] State Key Laboratory of Proteomics, Academy of Military Medical Sciences, Academy of Military Sciences, 100850 Beijing, China. [3] State Key Laboratory of Experimental Hematology, Institute of Hematology, Fifth Medical Center of Chinese PLA General Hospital, 100850 Beijing, China. [4] Key Laboratory for Regenerative Medicine of Ministry of Education, Institute of Hematology, School of Medicine, Jinan University, Guangzhou, China. [5] These authors contributed equally: Chen C. Li, Guangyu Zhang. ✉email: rainyblue_1999@126.com; bingliu17@yahoo.com; ahe@pku.edu.cn

Studying the cellular origin and molecular regulation of hematopoiesis are of paramount importance. There are three waves of hematopoiesis during mouse embryo development, with definitive long-term hematopoietic stem cells (LT-HSCs) arising last at the aorta–gonad–mesonephros (AGM) region[1–3]. LT-HSCs gradually emerge through the process of endothelial-to-hematopoietic transition (EHT), acquiring the ability of self-renewal and reconstruction of all blood lineages[4–6]. Aortic progenitor cells may differentiate into aortic endothelium or definitive hemogenic endothelium, revealed by genetic reporter studies in zebrafish and in vitro cell culture[7–9]. In murine, a subpopulation of early arterial endothelial cells (eAECs) has been identified to contribute to hemogenic endothelial cells (HECs), subsequently pre-hematopoietic stem cells (pre-HSCs) and finally LT-HSCs based on single-cell RNA-seq (scRNA-seq)[4,10–12]. Nevertheless, how and at which stage a selected fraction of endothelial cells take on the path of EHT remains unresolved.

A better understanding of gene regulation and cell fate decisions requires comprehensive knowledge of the spatial 3D genome organization and its influence on transcriptional output. During cell differentiation, the 3D genome structure is dynamically arranged at multiple scales, including large-scale compartments (A/B), topologically associating domains (TADs), and chromatin loops[13,14]. The 3D genome re-organization shapes the transcriptional landscape and vice versa to some extent at different scales. Some regulatory events are not required for TAD boundary formation or higher-level compartmentalization but are important for sub-TAD connections[15]. Apart from the structural proteins cohesin and CTCF, transcription factors (TFs), chromatin regulators, and histone modifications also play crucial roles in shaping multiscale genome architecture during development and diseases[16–18]. Direct E-P looping is one of the mechanisms, by which enhancers target promoters and control the gene transcriptional activity and specificity[19]. These interactions are predicted to most likely occur within TADs[20,21]. It has attracted great attention to study how these multi-layer regulatory inputs are hierarchically coordinated over time during cell fate specification.

Efforts have been focused on deciphering the molecular basis in HSC formation through EHT. A recent study compared 3D genome organization in fetal and adult HSCs, showing that large-scaled compartments and TADs are relatively conserved while intra-TAD interactions are much more dynamic and stage-specific[22]. Tan and colleagues also approached the dynamic transcriptional network underlying HSC ontogeny by performing RNA-seq and histone mark ChIP-seq on five cell populations from endothelium through fetal liver to adult LT-HSCs, nominating a few TFs with potential roles in HEC formations[23]. To examine which subsets of HECs undertake the differentiation trajectory to HSCs, they performed both scRNA-seq and scATAC-seq and reconstructed a continuous developmental path. An intermediate cell state termed pre-HEC with great fate potential to ultimately lead to HSCs was identified[24]. RUNX1 is a master TF implicated in many stages of hematopoiesis, including EHT in early embryos as well as leukemia development[6,25]. The germline deletion of *Runx1* resulted in the LT-HSC elimination and the embryonic lethality by E12.5[26,27]. RUNX1 together with other hematopoiesis-related TFs, GFI1, SPI1, and FOSB (FGRS), has been shown to in vitro reprogram endothelial cells into HSCs[28]. Several studies have further characterized RUNX1 binding target genes in cell cultures or bulk samples of adults[29–31]. However, thus far, none of them has examined how RUNX1 regulates gene transcription in the context of defined chromatin states in embryos. Further, an integrated knowledge of epigenomic configurations of the 3D genome structure, histone modifications, and TFs in transcriptional regulation to promote

EHT is still lacking. This is notably challenging in part due to the extremely low number of cells available in early embryos for ChIP-seq with histone marks and TFs as well as Hi-C in measuring chromatin interactions.

Here, we improved and used itChIP-seq (indexing and tagmentation-based ChIP-seq)[32] and sisHi-C (small-scale in situ Hi-C)[33] in low-input samples to profile histone modifications, RUNX1 binding, and the 3D genome, for integrated analysis with gene transcription. Different from tedious steps involving the sonication and ligation in traditional ChIP-seq, itChIP-seq adopts Tn5-transposase-based tagmentation as the strategy for simultaneous chromatin fragmentation and barcoded adapter ligation for the one-step PCR enrichment. itChIP-seq allows for highly sensitive profiling of histone modifications and TFs in low input samples as few as 100 cells[32]. Four purified populations, early AECs, HECs, pre-HSCs, and LT-HSCs, are based on combined immunophenotypes we recently defined in mouse embryos[4,34–36]. We quantify the dynamic trajectories in different scales of the 3D genome structure until the scale correlates well with dynamic histone modifications in biological processes. HSC regulatory elements are already pre-configured with active histone modifications as early as eAECs, preceding small-scale chromatin structure dynamics. Developmental stage-specific TFs are identified in partnering with RUNX1 to drive the formation of HSCs. Together, our data support a model in illustrating how temporal epigenomic configurations fulfill cell lineage specification during development.

## Results

**Large-scale 3D genome re-organization with chromatin modifications along HSC ontogeny**. To examine molecular underpinnings in the continuum of HSC development, early AECs, HECs, pre-HSCs, and LT-HSCs of mouse embryos were isolated from AGM region at embryonic day 10.0 (E10.0) to fetal liver at E14.5 (Fig. 1a, b and Supplementary Fig. 1a–c), as we recently defined[4,10]. The eAECs are termed with regard to lateAECs as we previously reported[4]. Thus, eAECs represent a specific subpopulation of early progenitors with differentiation potential toward HECs or lateAECs, verified by the previous CFU-C assay[4]. We performed Hi-C and itChIP-seq experiments to delineate the dynamic trend in the 3D genome re-organization in association with several feature histone modifications using 200–500 cells per experiment[32,33] (Supplementary Tables 1 and 2). At a large-scale view, only 10.78% of genomic regions displayed A/B compartment flips across the four populations (Fig. 1c).

We next measured the dynamics in the TAD boundaries. For simplicity, we defined the boundary strength as minus the insulation score, which was reported previously[37]. Having confirmed the reproducibility and data quality of ChIP-seq data (Supplementary Fig. 1d), we calculated the normalized enrichments of four feature histone modifications around TAD boundaries. Interestingly, H3K27ac (active enhancers) and H3K4me1 (primed enhancers) signals were markedly accumulated within TADs, while H3K4me3 (promoters) peaked at TAD boundary regions (Fig. 1d and Supplementary Fig. 1e). H3K27me3 signals (a repressive mark) were markedly lower at the TAD boundary and relatively accumulating within the TAD (Fig. 1d). It was in accordance with the past observation that most TAD boundary regions displayed H3K4me3 enrichments[38–41]. Using top 1000 variable TAD boundaries on four populations, we identified four main clusters with increased or decreased boundary strengths (Fig. 1e). Gene Ontology (GO) term enrichment analysis showed that TAD boundaries with increased strengths in cluster 1 (C1) were enriched for non-hematopoiesis related tissue morphogenesis and associated with fewer

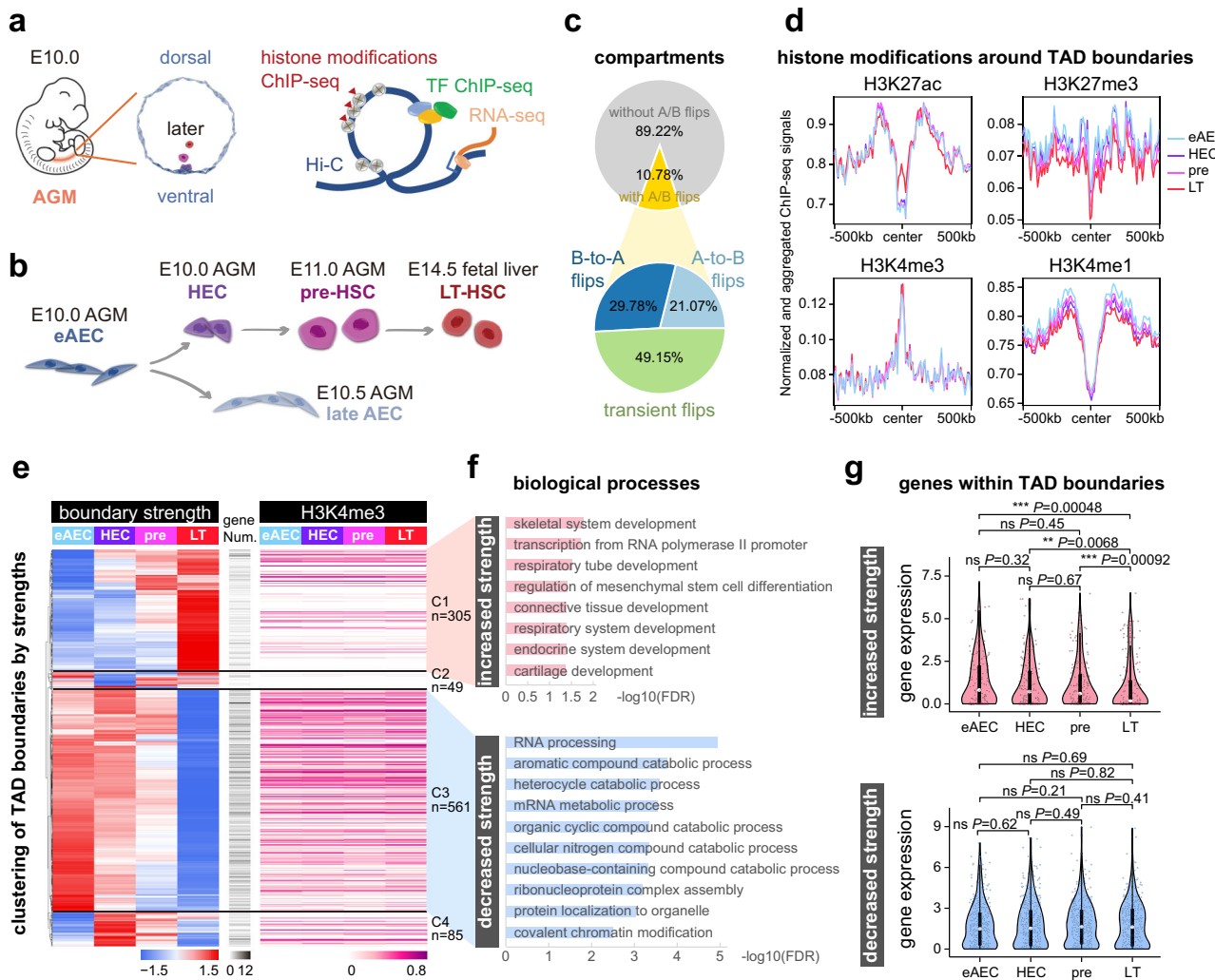

**Fig. 1 Annotating the large-scale 3D genome re-organization with chromatin states during HSC formation. a** Schematic of experimental design for multi-omics profiling across four populations. **b** The sampling scheme. Surface markers were provided in the methods. **c** Pie charts for the ratio of A/B compartment flips. B-to-A flips indicate a resulting compartment B to A change during differentiation, and A-to-B flips likewise indicate A to B; Transient flips indicate the change involving both A to B and B to A along the differentiation path. **d** Aggregate curves showing features of different histone modification ChIP signals around TAD boundary regions. The values were calculated with signals on ChIP-seq peaks. Each region was ±500 kb around TAD boundary centers, with 10 kb bin size. **e** Heatmap showing the clustering of top 1000 variable TAD boundaries (40 kb bin) by boundary strengths (row-scaled), and the corresponding H3K4me3 signals at TAD boundaries. Each row was one TAD boundary, identified by insulation scores. The minus value of the insulation score was boundary strength. The numbers of genes within each TAD boundary were shown. The ANOVA *P*-value (one-side) of H3K4me3 signals at TAD boundaries with increased strength among four stages was 0.922. The ANOVA *P*-value (one-side) of H3K4me3 signals at TAD boundaries with decreased strength among four stages was 0.353. **f** Gene ontology analysis for biological processes of TAD boundary clusters. The GO analysis was done by GREAT. *P*-value was calculated by two-sided binomial test and adjusted by Benjamini–Hochberg (BH) correction (FDR). The *X*-axis was −log10(FDR) of each term. **g** Violin plots of gene expression as in (**f**). Gene expression values were calculated and normalized with log2(TPM/10 + 1) from previous scRNA-seq data (GSE139389 and GSE67120). Each dot represented one single cell. In boundaries with increased strength, *n* = 166 genes. In boundaries with decreased strength, *n* = 429 genes. Box-and-whiskers plots represented the maxima, 75th percentile, median, 25th percentile, and minima. Pairwise comparisons between any two populations were performed through two-sample Wilcoxon Rank Sum test (two-side). **P*-value < 0.05; ***P*-value < 0.01; ****P*-value < 0.001; *****P*-value < 0.0001; "ns" not significant. eAEC early AEC, pre pre-HSC, LT LT-HSC.

H3K4me3-positive genes. Cluster 3 (C3) boundaries with decreased strengths were mainly linked to cellular catabolic processes and associated with more H3K4me3-positive genes (Fig. 1f). However, H3K4me3 enrichment levels at TAD boundaries were statistically stable across stages (Fig. 1e). Next, we clustered all identified TAD boundaries across four-cell populations and calculated associated H3K4me3 signals, further supporting the above finding (Supplementary Fig. 1f). These lines of evidence suggested that the TAD boundary strength might play a role implicated in HSC dormancy, as also reported elsewhere[42]. However, the genes on boundary clusters C1 and C3 did not

display a significant transcriptional change (Fig. 1g). Since metabolic pathways have been established to be involved in hematopoiesis across many stages[43], we reasoned that transcriptional alteration may take place later beyond this developmental window than these boundary strength changes as detected here. Together, these data support that the TAD boundary dynamics is associated with relatively stable expressions of boundary genes, less likely engaged in coordinating hematopoiesis. Despite of no detected alteration in transcription, examination of the TAD boundary strength dynamics may help predict future changes in metabolic pathways required for hematopoiesis.

**Connection dynamics between intra-TAD connectivity and feature histone modifications**. Lineage-specifying regulatory DNA elements, such as enhancers, are often labeled by a combination of histone modifications, such as H3K27ac, H3K27me3, H3K4me1, and H3K4me3. We next examined the dynamic interplay between intra-TAD connectivity and feature histone modifications. The domain scores of HECs, pre-HSCs and LT-HSCs were clustered together, with higher Pearson correlations. However, early AECs were well separated with lower correlations from three later populations, indicating a sharp transition in the 3D genome re-organization between early AECs and HECs (Supplementary Fig. 2a). Hierarchical clustering of all TADs by domain scores was made alongside with corresponding four histone modifications along with HSC development (Supplementary Fig. 2b). We identified 2068 significantly varying TADs with the cutoff FDR < 0.001 across four stages. Clustering of these 2068 TADs was used to identify their relevance to hematopoiesis and non-hematopoiesis processes, supporting the influence of intra-TAD interactions on EHT (Supplementary Fig. 2c–f). To establish the relationship between dynamic TADs and histone modifications, we identified five clusters based on top 1000 variable TADs (Fig. 2a). C1 with decreased TAD connectivity was accompanied by H3K27ac decrease from pre-HSCs to LT-HSCs (Fig. 2a–c and Supplementary Fig. 2g). Consistently, Gene Ontology (GO) analysis supported that these associated genes acquiring a silencing chromatin state were enriched for the pathways of multiple non-hematopoiesis tissue morphogenesis (Fig. 2d). C2 with a sharp decrease in TAD connectivity between eAECs and HECs displayed increasing H3K27me3 (high level) and H3K4me1 (low level), with nearly complete depletion of H3K27ac and H3K4me3 (Fig. 2a–c and Supplementary Fig. 2g). C3 with a gradually increasing TAD connectivity showed strong H3K27ac, H3K4me1, and H3K4me3 signals as early as eAECs (Fig. 2b, c). Specifically, H3K27ac increased significantly during the pre-HSC to LT-HSC transition, companied by a gradual reduction in H3K27me3 (Fig. 2c, e). Interestingly, C3 TAD regions were associated with several hematopoiesis-related functional terms (Fig. 2d). Therefore, these findings suggested that there was an enhanced intra-TAD connectivity associated with hematopoiesis during HSC development, with active chromatin states pre-established in the early stage.

We further looked into the temporal interplay between C3 TAD connectivity and chromatin states since they were closely associated with hematopoiesis. C3 TADs were primed with a certain amount of H3K27ac signals starting from eAECs, maintaining steadily through EHT and increasing in LT-HSCs (Fig. 2e and Supplementary Fig. 2h, i). In contrast, H3K27me3 manifested a reduction along the differentiation path (Fig. 2e and Supplementary Fig. 2h, i). Gene expression showed a delayed response compared with the activation of regulatory elements (Supplementary Fig. 2j). Most genes with increased expressions in C3 TADs were related to hematopoiesis and immune processes. These data indicated that important regulators of HSCs within C3 acquire enhancer activation at early AECs but become late activated upon the intra-TAD connectivity further enhanced over time.

Changes in intra-TAD contacts within short-range (≤2 MB) and long-range contacts (2–10 MB) may indicate differential regulatory mechanisms. Our genome-wide analysis presented significantly up-regulated short-range intra-TADs contacts in HECs, compared to eAECs (Fig. 2f). We also noted that some long-range contacts also gradually emerged (Supplementary Fig. 3a), despite a small portion compared to that in the short-range scale. RUNX1 is a pivotal TF required for EHT and HSC formation. Specific inspection and quantification of interaction strengths and histone modifications at the *Runx1* exemplified the

increased intra-TAD contacts (Supplementary Fig. 3b–d). The *Runx1* P1 promoter interacted strongly with two neighboring regions, as quantified by the observed/expected value (Fig. 2g, h). This identified the newly engaged *Runx1* P1 promoter in HECs evidenced by H3K4me3 signals and the enhancer activation by H3K27ac signals (Fig. 2g).

Taken together, our data indicated that HSC-activated regulatory elements are primed with active histone modifications at early AECs, and acquire high chromatin contacts to locate cognate target genes during HSC development.

**Pre-configuration with histone modifications at chromatin loops during HSC development**. We further investigated chromatin loops at the finer structural scale (Supplementary Fig. 4a, b). Most chromatin loop anchors were distributed at enhancers and promoters (Supplementary Fig. 4c). We thus focused on loops involving enhancers or promoters, including E–E, E–P, and P–P loops (Supplementary Fig. 4d). The normalized counts per million mapped paired reads (CPM) were used to quantify loop strengths. Based on top 1000 variable loops involving enhancers or promoters, they were classified into three clusters (Fig. 3a). In line with the finding above, H3K4me3 and H3K27ac signals and gene expression asynchronously changed during development (Fig. 3b, c). Further functional annotation of these genes showed that C1 with decreased loop strengths were enriched for non-hematopoiesis, C2 with a sharp increase during HECs to pre-HECs transition for immune-related terms, and C3 with a sharp increase during early AECs to HECs transition for several hematopoiesis related terms (Fig. 3d and Supplementary Fig. 4e). Interestingly, the decrease in loop strengths in C1 was associated with strong H3K27ac signals from early AECs to pre-HSCs but to a lesser degree in LT-HSCs. Similarly, the increase in loop strengths in C2 or C3 was associated with strong H3K27ac signals but to a higher degree in LT-HSCs. Next, we focused on specific gene loci related to hematopoiesis or endothelium within these clusters (Fig. 3e, f). At *Ets1*, an EC marker gene in C1 cluster, loop loss over time was accompanied by a reduced enhancer activity and subsequent down-regulated gene expression. At *Cebpg*, an immune marker gene in C2 cluster, the gain of loops was followed by increased enhancer activity. At *Meis1*, a hematopoiesis marker gene, increasingly emerged loops were observed in association with a slight increase in the enhancer activity and a marked increase in gene expression (Fig. 3e, f). These results suggested that HSC-related chromatin domains are pre-configured with active histone modifications while accompanied by furthering dynamic chromatin loop strengths.

**Priming interactions by RUNX1 in enhancer–promoter loops**. Loop strengths themselves could not predict transcription states, which are concurrently influenced by chromatin states and chromatin binding regulators, such as lineage-specifying TFs. Thus, we attempted to identify TFs with a potential role in mediating such loop formations and transcriptional activation in C3. Expectedly, CTCF was the most significant TF among of them. Importantly, several interesting known TFs were also over-represented, including RUNX1. Consistently, *Runx1* expression was also significantly higher in later stages than early AECs, while *Hif2a* expression decreased (Fig. 4a).

RUNX1 is as a well-known master TF essential for developmental hematopoiesis in orchestrating the EHT process[44,45]. We examined its role in enhancer and promoter looping by performing RUNX1 itChIP-seq using as few as 500 cells (Supplementary Table 3) for integrated analysis with Hi-C data. We asked to what extent RUNX1 is involved in such loops between enhancers and promoters during HSC development.

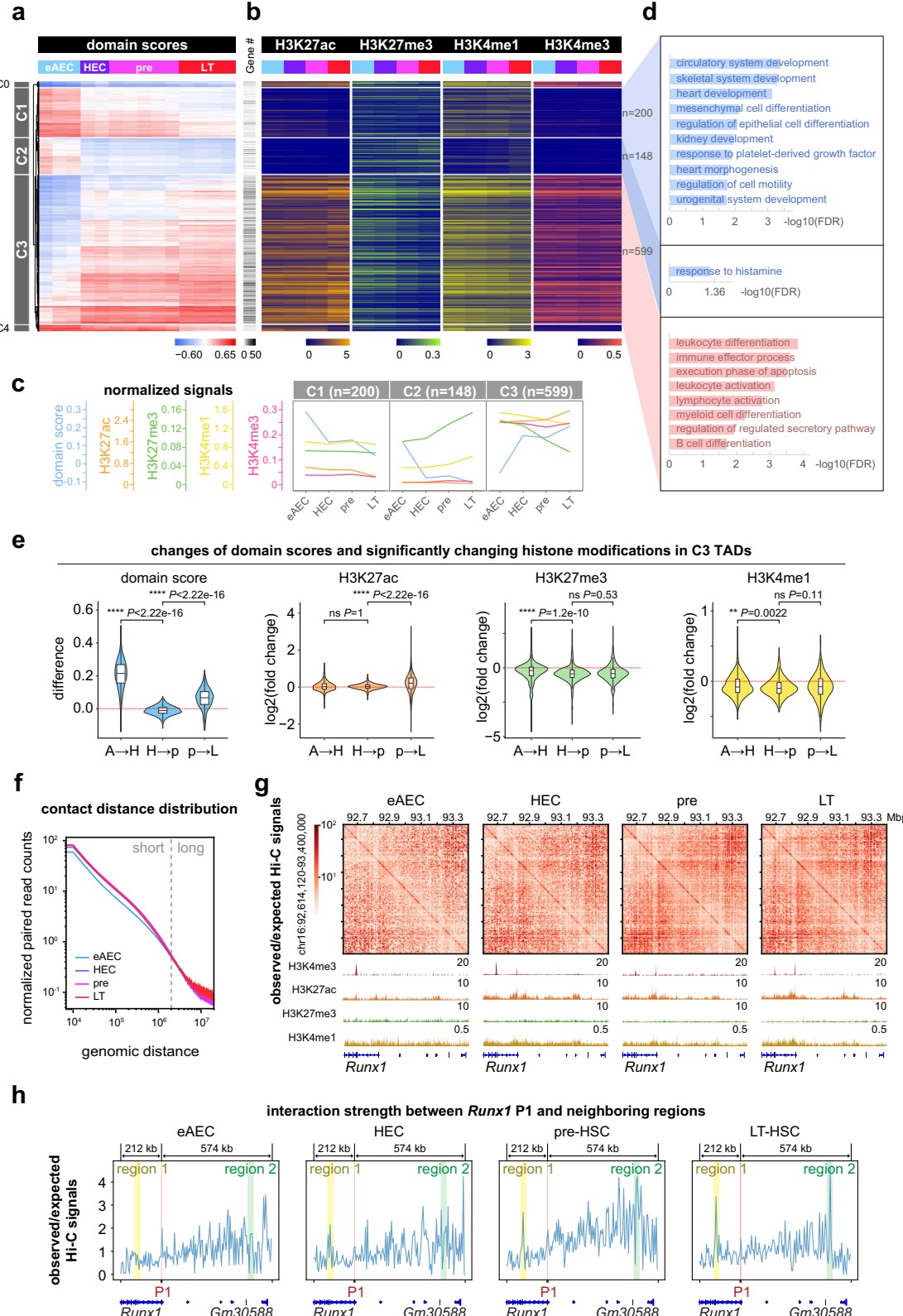

Surprisingly, in addition to RUNX1 peaks identified in HECs, pre-HSCs and LT-HSCs, we also found a considerable number of highly reproducible RUNX1 peaks in eAECs, which are previously considered to have a very limited expression of RUNX1. To further exclude the plausible eAEC "peaks" due to background, we analyzed our RUNX1 datasets with other public RUNX1-ChIP-seq but from other cell types at specific regions (Supplementary Fig. 5a) and genome wide (Supplementary Fig. 5b). This supports that RUNX1 peaks identified in our data are biologically meaningful and highly enriched for the RUNX1 motif. Further, we sought to examine the intersection between anchor regions of enhancers and promoters and RUNX1-binding sites across four populations, separating into E–E, P–P, and E–P interactions with or without RUNX1 peaks. Strikingly, 40.9% of

**Fig. 2 Mapping developmental dynamics of intra-TAD connectivity and chromatin configuration of feature histone modifications. a** Heatmap showing clusters of the top 1000 variable TADs by normalized domain scores. **b** Heatmap of H3K27ac, H3K27me3, H3K4me1, and H3K4me3 signals in corresponding ChIP-seq peaks within TADs as sorted in (**a**). **c** Quantification curves for the asynchronous change of feature histone modifications within TADs and TAD domain scores in three main clusters. Mean values were calculated for TADs in each cluster. **d** Biological processes enriched in three main clusters of TADs. The GO analysis was done by GREAT. *P*-value was calculated by two-sided binomial test and adjusted by BH correction (FDR). **e** Violin plots showing an asynchronous change of TAD domain scores and histone modifications in C3 TADs. The log2(fold change) values between two adjacent stages were calculated. Box-and-whiskers plots represented the maxima, 75th percentile, median, 25th percentile, and minima. The two-sample Wilcoxon Rank Sum test (two-sided) was performed for C3 TADs ($n = 599$) between two adjacent stages. A early AEC, H HEC, p pre-HSC, L LT-HSC. **f** Accumulative curves showing the distribution of normalized paired-read counts in Hi-C at 10 kb matrices within different genomic distances. **g** Exemplification showing the increasing intra-TAD interactions with observed/expected Hi-C matrices (5 kb resolution) at *Runx1* gene site, associated with feature histone modifications. **h** Curves quantifying the changing loop strength between the *Runx1* P1 promoter and neighboring regions. The genomic coordinate is chr16:92,614,120–93,400,000, from −212 kb to +574 kb relative to *Runx1* P1. The observed/expected Hi-C matrices at 5 kb resolution were used for calculation. The *Runx1* P1 region in red was TSS ± 2.5 kb. The region 1 in yellow overlapped with *Runx1* P2 promoter, while the region 2 in green represented a distal enhancer. *\**P*-value < 0.05, \*\**P*-value < 0.01, \*\*\**P*-value < 0.001, \*\*\*\**P*-value < 0.0001, "ns" not significant. eAEC early AEC, pre pre-HSC, LT LT-HSC.

---

E–P interactions were bound by RUNX1 (Fig. 4b). Clustering of 1462 genes with RUNX1-engaged E–P by gene expression yielded four clusters. C1 genes showed the highest expression in pre-HSCs. C2 genes showed the steadily increasing expression, accompanied by a marked increase in RUNX1 binding signals, H3K27ac and H3K4me3 in LT-HSCs and a slow reduction in H3K27me3. These C2 genes were associated with hematopoiesis and immune related functional terms (Fig. 4c–e). C3 genes showed high expression in HECs. C4 genes showed a gradually decreasing expression, accompanied by overall decreased RUNX1 binding signals and associated with non-hematopoiesis terms, such as vasculature and heart development (Fig. 4c–e).

To further examine the relationship between RUNX1-engaged E–P interactions and its potential function in hematopoiesis, we focused on C2 cluster, enriched for the GO terms, hematopoiesis, and immune. Of C2 genes, several were known to be related to definitive hematopoiesis, while others have been reported to participate in the terminal differentiation of LT-HSCs into other lineages. *Zfp36l2* functions in definitive hematopoiesis[46,47]. *Fbxo11*, *Msh2*, *Eml4,* and *Tgif1* play important roles for B cell, T-helper cell, and erythroid cell differentiation, respectively[48–51]. *Hnrnpll* encodes an RNA-binding protein required in alternative splicing of *Ptprc* and *Stat5a*[52,53]. *Cdc42ep3* and *Eml4* are related to microtubules[54]. These genes already manifested evident interactions in early AECs, further strengthened in HECs and pre-HSCs, with the exception of a few interactions becoming weaker in LT-HSCs. Sub-clustering by the interaction strengths showed that RUNX1-engaged E–P interactions with decreased strengths were linked to mitotic cell cycle and angiogenesis (Fig. 4f and Supplementary Fig. 6a). The interaction strengths within the gene loci enriched for the GO terms, B cell proliferation and interferon α/β signaling, appeared transiently higher in HECs and pre-HSCs (Fig. 4f and Supplementary Fig. 6a). Interestingly, the increasing sub-cluster was related to T cell activation, adaptive immune, and definitive hemopoiesis (Fig. 4f and Supplementary Fig. 6a). Statistical analyses were also performed (Fig. 4g).

We also investigated RUNX1 occupancy in the C4 cluster. Unsupervised subclustering of C4 nominated four subclusters (Supplementary Fig. 6b). The changes in H3K4me3 and H3K27ac signals coincided with RUNX1 binding. Surprisingly, the loops associated with decreased RUNX1 signals were well-correlated with gene expression, but not in those with increasing RUNX1 signals, indicating other repressive mechanisms in operation. C4 genes with decreased RUNX1, H3K4me3, and H3K27ac signals that were mainly linked to biological processes, such as endothelial cells or other non-hematopoiesis tissues (Supplementary Fig. 6c).

Taken together, our results indicated RUNX1-engaged E–P interactions promoted hematopoiesis early as in eAECs, which were reinforced within EHT. Presumably, RUNX1-engaged E–P interactions might participate in suppressing non-hematopoiesis gene programs, but not further explored in this study.

**RUNX1 co-TFs in mediating central HSC promoter–enhancer loops.** RUNX1 has been shown with the ability to establish de novo target sites, initiating the local increase in active histone modifications at distal *cis*-regulatory elements in in vitro HEC differentiation[55]. To precisely define the co-TFs required to initiate EHT and HSC formation, we examined the potential enriched TF motifs in RUNX1-engaged E–P interaction regions. Among the anchor regions of RUNX1-engaged E–P interactions, we selected distal enhancer regions (intergenic H3K27ac peaks excluding TSS ± 5 kb regions) and promoter regions (within TSS ± 5 kb). Most RUNX1 peaks were distributed in proximal promoters (Fig. 5a and Supplementary Fig. 7a). Only nearly 10% of distal enhancers from the interactions were bound by RUNX1 in four populations. Proximately 40–50% of promoters from the interactions are bound by RUNX1, slightly higher in LT-HSCs, and likely positively correlated with the *Runx1* expression level.

RUNX1-engaged E–P interactions were separated into three types, with RUNX1 occupancy on promoters, enhancers or both (Fig. 5b). First, we identified potential co-TF candidates within enhancers and promoters of RUNX1-engaged E–P interactions separately (Fig. 5b). It was noted that enhancers or promoters with RUNX1 occupancy showed a higher significance of TF motif enrichment than those without RUNX1 (Supplementary Fig. 7b, c). The E–P interactions with RUNX1 occupancy on both, enhancers or promoters presented indistinguishable numbers of significantly enriched TF motifs (Supplementary Fig. 7b).

We further confined the list of potential co-TFs by integrating gene expression particularly for the TF families with the identical binding consensus sequences (motif matrix). We focused on a few top significant TF families whose expression was also significant in either population or in an increasing trend (Fig. 5d). TFs identified in E–P interactions without RUNX1 binding were used as the control (Supplementary Fig. 7c). We identified not only known co-TFs, such as GFI1b, PU.1, and GATA2, but also many co-TFs with potential roles in EHT and HSC formations, including EGR1, ATF1/2/4, NRF1/2, IRF1/3/8, SMAD4, and KLF6 (Fig. 5d), although further validation in vivo is demanded. In summary, these data indicated that RUNX1 proteins occupy target E–P interactions primed with feature active histone marks before EHT. It is tempting to speculate that RUNX1 and co-TFs identified within E–P interactions might function as a TF module

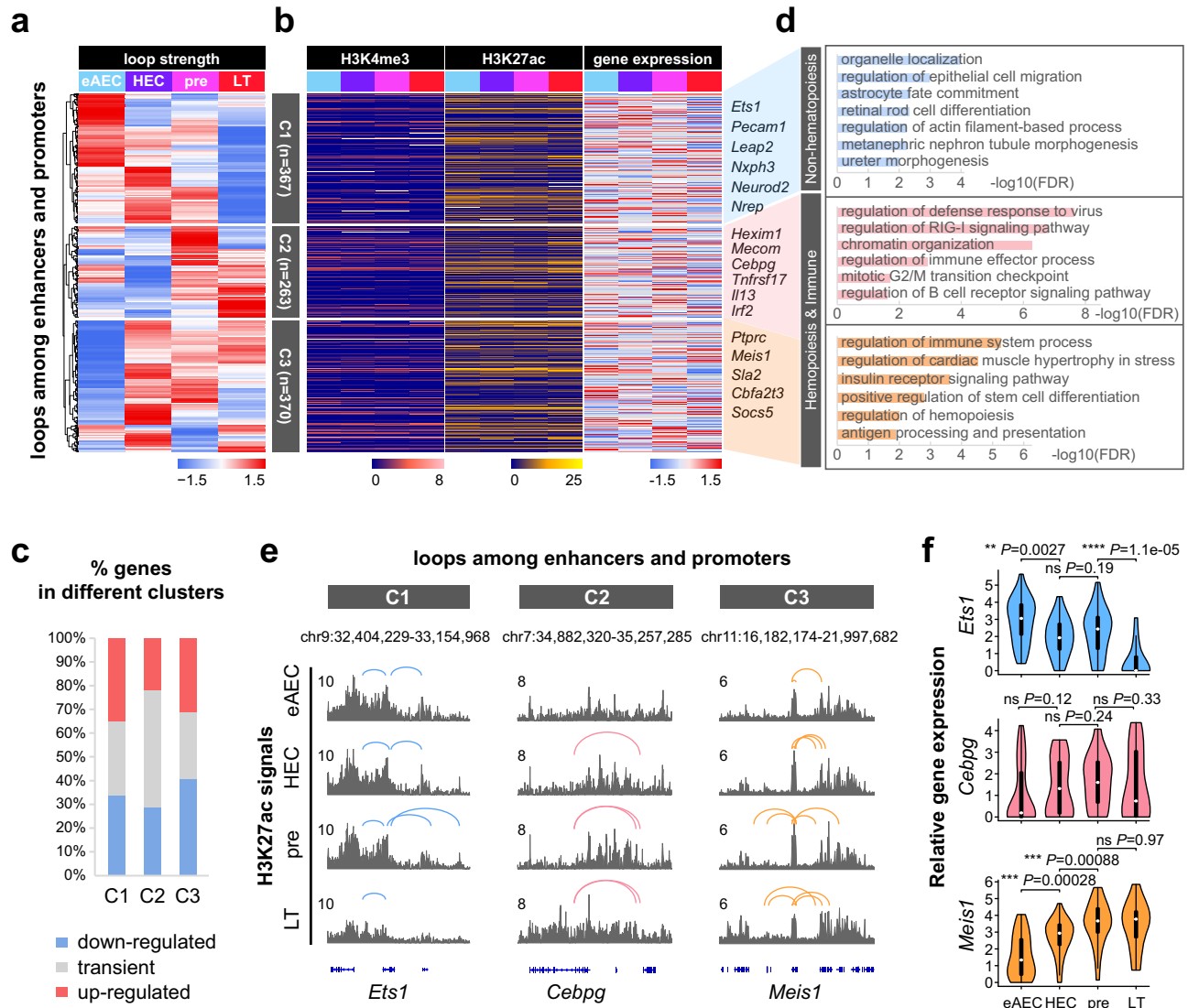

**Fig. 3 Pre-configuration with feature histone modifications precedes chromatin loop dynamics. a** Heatmap showing clusters of top 1000 variable loops among enhancers and promoters by loop strengths (normalized CPM). Loops contained E–E, E–P, or P–P loops. **b** Heatmaps showing cognate H3K4me3, H3K27ac, and gene expression on anchors of each loop. Only reads on ChIP-seq peaks were used to calculate normalized signals of H3K4me3 and H3K27ac. The eAEC and HEC scRNA-seq data were from GSE139389. Pre-HSC and LT-HSC scRNA-seq data were from GSE67120 and GSE66954. Gene expression values were calculated and normalized with log2(TPM/10 + 1). **c** Percentages of genes in three loop clusters with down-regulated, transient, and up-regulated expression from eAECs to LT-HSCs. Transient genes meant those whose expression was up-regulated in some cases and down-regulated in others. **d** Biological process analysis of looping regions in three clusters. The inputs were these looping anchor regions for GO term analysis with GREAT. The −log10(FDR) values were shown after BH correction. **e** Track views showing the loop change trend among enhancers and promoters at the Ets1, Cebpg, and Meis1 loci. **f** Violin plots showing expression of representative genes in three clusters with log2(TPM/10 + 1). The scRNA-seq data contain 42 eAECs, 33 HECs, 50 pre-HSCs and 16 LT-HSCs. Box-and-whiskers plots represented the maxima, 75th percentile, median, 25th percentile, and minima. Wilcoxon Rank Sum test (two-sided) was performed between two adjacent stages. *P-value < 0.05, **P-value < 0.01, ***P-value < 0.001, ****P-value < 0.0001, "ns" not significant. eAEC early AEC, pre pre-HSC, LT LT-HSC.

to license the transcriptional specificity of targets, pre-configured with chromatin architecture and histone modifications (Fig. 5e).

## Discussion

In summary, by integrating multi-omics datasets generated from Hi-C and itChIP-seq for histone modifications and RUNX1 as well as scRNA-seq in low-input newly defined cell populations, we approached how the dynamics in multi-layered chromatin states shaped the path of HSC ontogeny. In particular, we followed the continuum of HSC formation starting from a population of early AECs with a small fraction expressing RUNX1, defined by a combination of cell surface markers[4,11,12,24,56].

Our study provides a biological annotation of the dynamic 3D genome structure in combination with histone modifications, TF binding, and transcriptome during early HSC development. We observed the dynamic TAD boundaries on large scale, but unlikely associated with hematopoiesis. Instead, intra-TAD dynamics were largely linked to HSC development. By focusing on differential enhancer–promoter interactions within these TADs, we found that about 40.9% of E–P interactions were bound by RUNX1 along with the HSC development. Further analysis identified both known and unknown TFs as RUNX1 interacting regulators to enhance hematopoiesis-related interactions. Importantly, our data support that HSC-specific chromatin loops

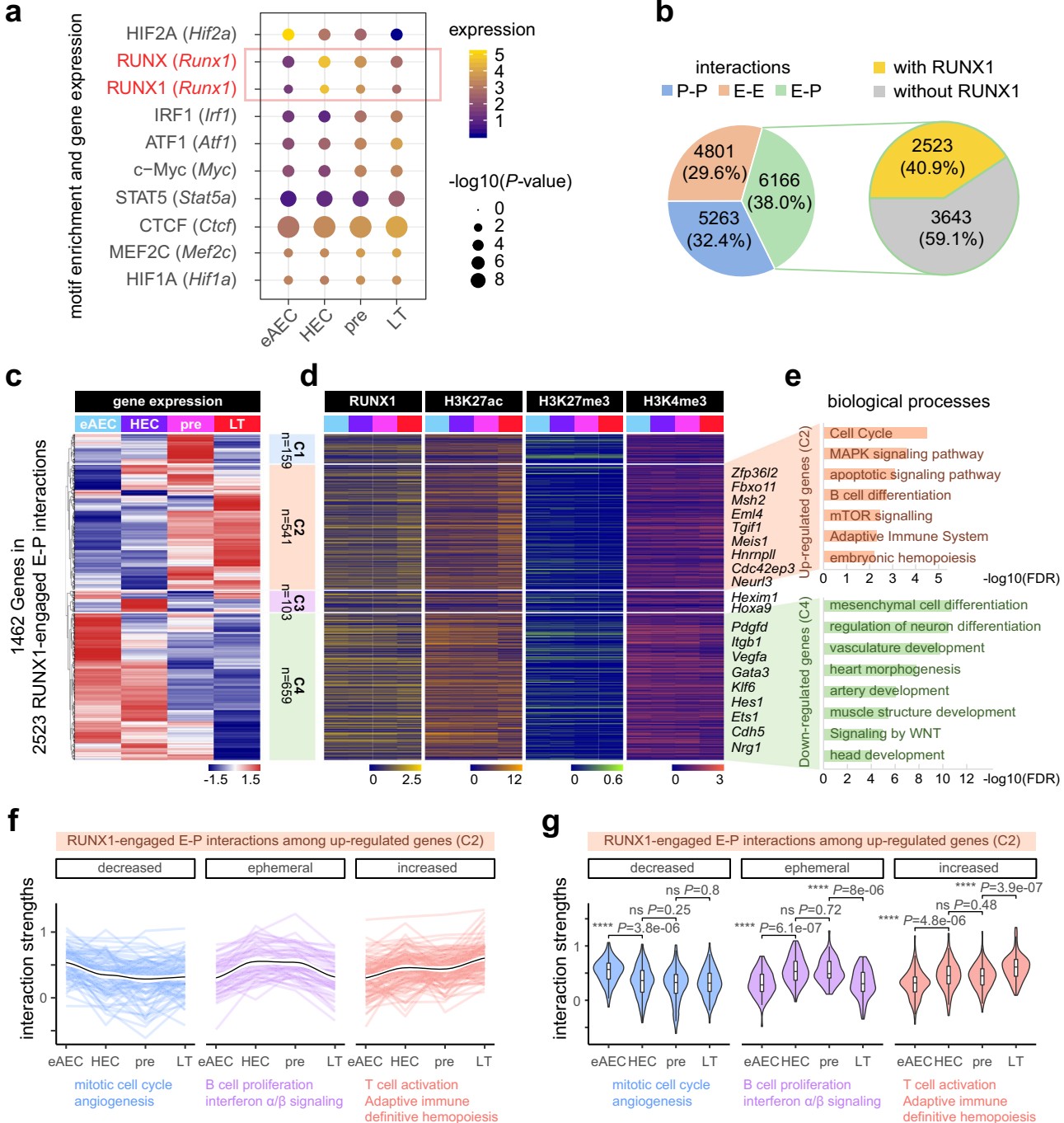

**Fig. 4 RUNX1 is engaged in looping between promoters and enhancers for priming genome interactions during HSC formation. a** Bubble plots showing gene expression and TF motif enrichment, identified at anchor regions of C3 loop clusters as in Fig. 3a, with most variable gene expression. TF motif analysis was performed by HOMER. *P*-value was calculated by two-sided binomial test. **b** Pie chart showing the portion of E–E, P–P, and E–P interactions with or without RUNX1 occupancy. **c** Heatmap showing hierarchical clustering based on the expression of genes with RUNX1-engaged E–P interactions. **d** Heatmap showing the corresponding RUNX1 binding and histone modifications on looping sites as in (**b**). Representative genes were listed on the right. The ANOVA *P*-values (one-sided) of TF or histone modification signals in C2 hematopoietic cluster were 1.27e−05 **** for RUNX1, 3.92e−12 **** for H3K27ac, 0.000216 *** for H3K27me3, and 0.0606 ns for H3K4me3. **e** Gene Ontology analysis of biological processes supporting the clustering. **f** Lines reflecting strengths of three sub-clusters of RUNX1-engaged E–P interactions among C2 genes. Those RUNX1-engaged E–P interactions existing from eAEC to LT-HSC were used for clustering. Smoothly fitted interactions dynamics into a function (method = 'loess') in each cluster, the black line indicates the predicted value. The thick white bars represented a 95% confidence interval. **g** Violin plots quantifying the strengths of three sub-clusters of RUNX1-engaged E–P interactions among C2 genes, corresponding to (**f**). For decreased interactions, *n* = 91. For ephemeral interactions, *n* = 60. For increased interactions, *n* = 113. Box-and-whiskers plots represented the maxima, 75th percentile, median, 25th percentile, and minima. The two-sample Wilcoxon Rank Sum test (two-sided) was performed between two adjacent stages. *P-value < 0.05; **P-value < 0.01; ***P-value < 0.001; ****P-value < 0.0001; "ns" not significant. eAEC early AEC, pre pre-HSC, LT LT-HSC.

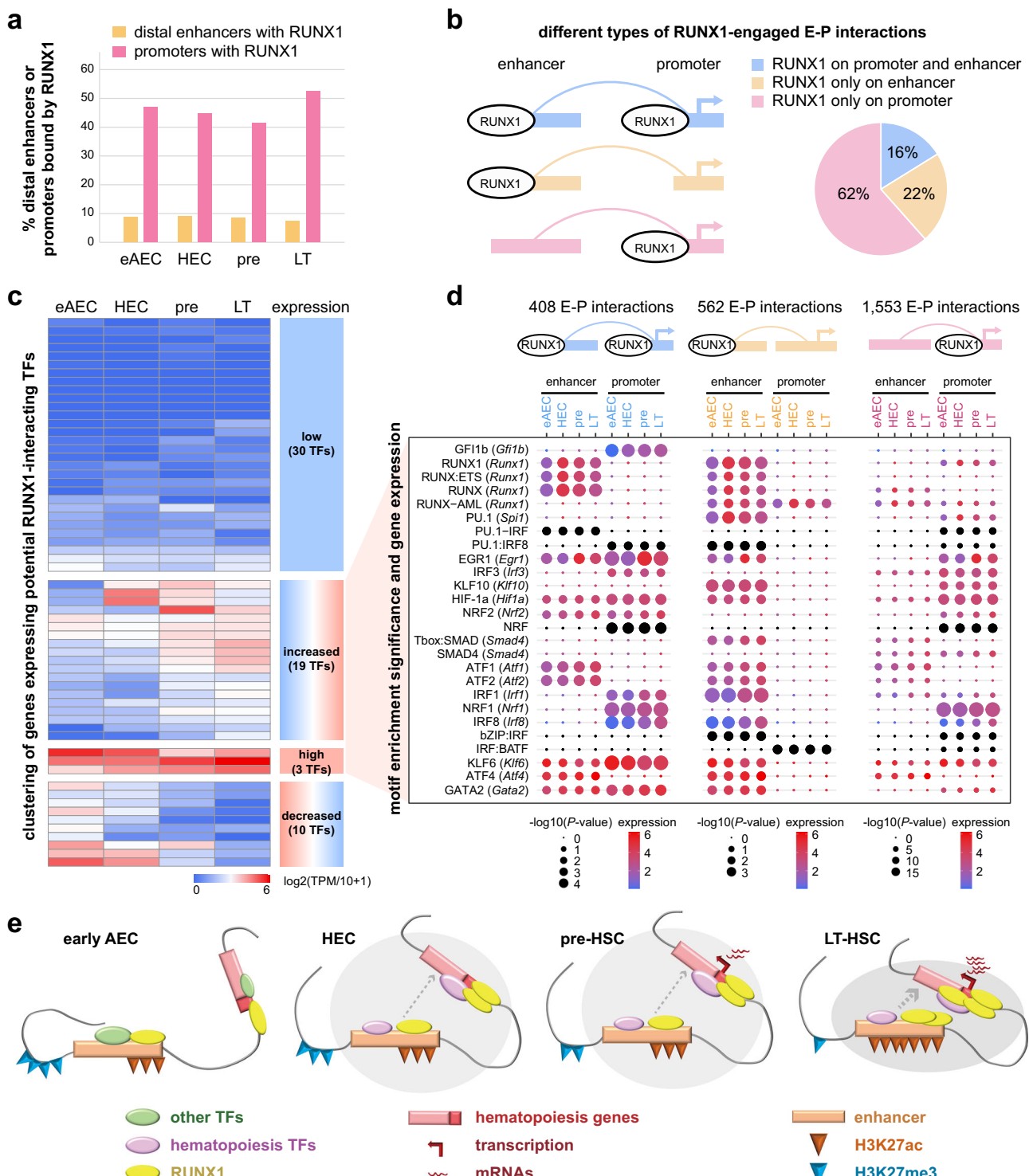

**Fig. 5 Identifying potential RUNX1 co-TFs in mediating enhancer–promoter interactions during HSC formation. a** Bar plot showing the portion of distal enhancers (with genomic distance to TSS > 5 kb) and promoters (with genomic distance to TSS ≤ 5 kb) within RUNX1-engaged E–P interactions. **b** Pie chart showing the portion of E–P interactions with RUNX1 occupancy on promoters, enhancers, or both. Schematic on the left shows the three types of genomic locations of RUNX1 occupancy. **c** Hierarchical clustering based on gene expression of potential RUNX1-interacting TFs identified through de novo TF motif discovery on the E–P looping regions as above. TF genes used for clustering were genes in families whose protein binding motifs were top 11 significant in all candidates identified in Supplementary Fig. 7b. Gene expressions were calculated by log2(TPM/10 + 1) from scRNA-seq data. **d** Bubble plots showing significances of TF motif enrichments as in (**c**), together with gene expression. TF motif analysis was performed by HOMER. *P*-value was calculated by two-sided binomial test. **e** Working model of multi-omics dynamics during HSC formation. RUNX1 occupies target promoters or/and enhancers primed with H3K27ac in eAECs. Repressive H3K27me3 gradually diminishes accompanying later H3K27ac increase. Other hematopoiesis related TFs are recruited to form a TF module with RUNX1 to facilitate enhancer–promoter interactions driving the core HSC gene expression. eAEC early AEC, pre pre-HSC, LT LT-HSC.

are stepwise reinforced while active histone modifications among these regions are pre-established before EHT and later enhanced in LT-HSCs. The pre-configured active histone modifications, H3K27ac and H3K4me1 as reporting enhancer activation, are likely attributed to the arterial program with the CD44[+] immunophenotype[4,24,57,58] in eAECs and HECs. This indicates that early AECs may be already specified to differentiate toward HECs and ultimately HSCs, likely explaining that the change in the large-scale 3D genome re-organization is not directly linked to HSC development-related genes. However, we concur that the clear differences observed between E10.0 eAECs and the E10.0/E11.0 EHT populations may not solely reflect an important transition in EHT, but also be influenced by at least part of the E10.0 eAECs being on a different fate trajectory from the E10.0 HE/pre-HSCs. Nevertheless, the identities of HECs and eAECs can be readily distinguished by their distinct transcriptome and domain score features, indicating the main component in each population. This finding still provides the epigenetic basis to delineate cell fates as previously inferred by the transcriptome from scRNA-seq[4,11,56].

Rather than focusing on the identification of stage-specific enhancers, we examined how spatial interactions in HSC are established between enhancers and their target genes, mediated by TFs. A recent report focused on enhancer dynamics across from RUNX1[−]CD31[+] ECs to bone marrow HSCs[23]. However, computational prediction of their target genes can be inevitably inaccurate to define the molecular basis of HSC ontogeny. Our work capitalized on the high-resolution Hi-C data to firstly determine the dynamic genome interaction associated with hematopoiesis enriched functional terms. This set of chromatin loops offers the high-precision map to understand how RUNX1 and co-TFs identified cooperate to guide HSC development. It may also inform the better design to reprogram or differentiate embryonic stem cells to HSCs for regenerative medicine.

RUNX1 is essential for both primitive and definitive hematopoietic development. To date, all studies profiled RUNX1-binding sites by ChIP-seq using a large number of cells from either in vitro cell culture or immortalized cell lines derived from patients[29–31,45,59,60]. Our study presents a time-course mapping of genome-wide RUNX1 occupancy using low-input itChIP-seq[32,33] in rare and purified cell populations around EHT in mouse embryos. Most RUNX1 binding sites in HSCs were found to be largely evident in early AECs. Strikingly, early AECs purified by CD31[+]CD44[+] already exhibited RUNX1-binding signals. Although this is in accordance with increasing *Runx1* expression during development, we cannot rule out the relative heterogeneity in early AECs, a mixed population of RUNX1[−] and RUNX1[+] corresponding to a small number of HECs in eAECs purified by the current FACS gating strategy. Further single-cell ChIP-seq for RUNX1 would be crucial to study its molecular functions during cell fate decisions, especially before and at the start of EHT. Notwithstanding this limitation, our work not only provides a rich source for the RUNX1-binding dynamics but also supports the prerequisite of RUNX1 chromatin engagement for HSC formation.

The integrated multi-omics dissection of HSC development in mouse embryos using relatively purified cell populations reveals that the pre-configured chromatin modifications are established prior to the formation of the mature 3D genome structure in HSCs. This finding informs a reprogramming strategy to derive cell types of interest from epigenetically neighboring cell lineages by manipulating key chromatin interactions as designed. Together with a previously established combination of TFs in an attempt to reprogram ECs to acquire HSCs, we are closing to the successful era of generating off-the-shelf HSCs for therapeutic treatments of blood diseases.

## Methods

**Mice**. All animal experiments were performed according to protocols approved by the Institutional Animal Care and Use Committees of Peking University and Academy of Military Medical Sciences. All animal experiments were approved by the Animal Care and Use Committee of the Institute. All cell populations were collected from fetuses of 8–10 weeks old wild-type female mice maintained on C57BL/6 background. Early embryos were staged by somite pair (sp) countings, such as 31–35 sp for E10.0 and 41–45 sp for E11.0. E14.5 embryos were staged by morphologic features. AGM region was dissected as reported[61]. Primary embryonic single-cell suspension was acquired using the type I collagenase for dissociation at 37 °C and PBS/5% FBS for washes.

**Flow cytometry**. The dissociated cells were incubated with antibody madoptixture at 4 °C for 30 min, followed by incubation with 7-AAD antibody at room temperature for 5 min. Cells were sorted and analyzed by flow cytometers FACS Aria II (BD Biosciences) and MoFlo XDP (Beckman Coulter) in the purity model. The FACS data were analyzed with FlowJo software (v10, Tree star). Surface markers for E10.0 early AECs in AGM regions were CD41[−]CD43[−]CD45[−]CD31[+]CD44[+]Kit[−]. Surface markers for E10.0 HECs in AGM regions were CD41[−]CD43[−]CD45[−]CD31[+]CD44[+]Kit[+]CD201[+]. Surface markers of E11.0 AGM pre-HSCs were CD31[+]Kit[+]CD201[+]. Surface markers for E14.5 fetal liver LT-HSCs were CD45[+]CD201[+]CD150[+]CD48[−]. Cells were stained using the following antibodies (1:100 diluted): Anti-CD31 (BD, MEC13.3, Catalog# 562939), Anti-CD41 (BD, MWReg30, Catalog# 553848), Anti-CD43 (BD, S7, Catalog# 553270), Anti-CD44 (BioLegend, Catalog# 103044), Anti-CD45 (BD, 30-F11, Catalog# 553079), Anti-CD48 (BioLegend, Catalog# 103432), Anti-CD201 (eBioscience, eBio1560, Catalog# 17-2012-82), Anti-Kit (eBioscience, 2B8, Catalog# 14-1171-82), and Anti-CD150 (BioLegend, Cat# 115904). 7-aminoactinomycin D (7-AAD; eBioscience, Catalog# 00-6993-50) was used to exclude dead cells. The FACS Diva 8 "index sorting" function was activated and sorting was performed in the single-cell mode.

**itChIP-seq**. The itChIP-seq[32] was performed with a few modifications. Briefly, after collection by FACS, cells were fixed by 1% formaldehyde solution at room temperature for 3 min followed by 1 × PBS washing and centrifugation at 4 °C. Samples could be stored at −80 °C or used immediately. Fixed cells were incubated in hypotonic buffer at 62 °C for 10 min (histone modification ChIP) or 37 °C for 1 h (TF ChIP), to relax chromatin. Cells were slightly sheared and then quenched with Triton X-100. Chromatin was digested by Tn5 assembled with MEA/B adapters at 37 °C for 1 h. After tagmentation reaction was stopped, the sample was sheared to release chromatin out of nuclei. By centrifugation at 4 °C, the soluble supernatant was collected and used for incubation with antibodies overnight. Dynabeads protein A (Invitrogen, 10001D, 30 mg/ml) beads were used to pull down chromatin fragments coupled with antibodies. Adapter-ligated fragments were eluted from beads and treated with proteinase K. DNA was purified and extracted by phenol–chloroform, then used for library preparation with Q5 polymerase (NEB) and Illumina Nextera index primers. After the size selection for 200–1000 bp fragments, libraries were quantified with Qubit for concentrations. Samples were mixed and sequenced on Nova 6000 (Illumina) for paired-end 150 bp reads. The itChIP-seq experiments were performed with the following antibodies: Anti-H3K4me3 (Millipore, Catalog# 04-745), Anti-H3K27ac (Diagenode, Catalog# C15410196), Anti-H3K27me3 (Millipore, Catalog# 07-449), Anti-H3K4me1 (Abcam, Catalog# ab8895), and Anti-RUNX1 (Abcam, Catalog# ab23980).

**sisHi-C**. The generation of sisHi-C library was performed as reported with a few modifications[33]. Different cell populations were sorted by FACS and fixed with 1% formaldehyde solution at room temperature for 10 min. *Mbo*I enzyme (NEB, R0147M) was used to digest chromatin. After biotinylation, ligation, and proteinase K treatment, DNA was purified and sheared to 300–500 bp with Covaris M220, followed by Dynabeads Myone Streptavidin C1 bead pulling down. The Hi-C libraries were amplified for 12–15 cycles with Extaq (Takara) and Illumina Nextera index primers. Size selection was carried out by first 0.5× AMPure beads to remove >1 kb fragments, and second 0.5× AMPure beads to the supernatant to obtain 200–1000 bp fragments for sequencing. Libraries were then quantified and sequenced with paired-end 150 bp reads on NovaSeq 6000 platform (Illumina).

**ChIP data processing**. The quality of sequencing was evaluated by fastqc (v.0.11.8). Raw reads were trimmed and removed for adapter sequences by cutadapt (v.2.6) with parameters "-q 20 -O 10 –trim-n -m 30 –max-n 0.1". Trimmed reads were mapped to mm10 mouse genome by Bowtie2 (v.2.3.5) with the parameter "-N 1". Mapped reads with MAPQ ≥ 30 were considered as uniquely mapped reads, sorted with samtools (v.1.9) for the subsequent analyses. Duplicates were removed with picard MarkDuplicates (v.2.2.4). Only non-duplicated reads were used for further analysis. Reads of replicates were sampled to the same level for comparisons by samtools (v.1.9). Bigwig files were generated with bamCoverage (v.3.3.1), normalized to 1M reads. Peaks were called with non-duplicated bam files by MACS2 callpeak (v.2.2.5).

**Correlation analysis**. To evaluate the correlations between replicates of itChIP data, we calculated Pearson correlation coefficients with normalized average scores in global bins or peak regions by multiBigwigSummary (v.3.3.1) from deepTools software[62] in "bins" or "BED-file" modes. Correlation heatmaps were generated by plotCorrelation in deepTools software.

**ChIP-enrichment calculation**. Non-duplicated reads on ChIP-seq peaks were normalized and used for calculation of histone modification or TF signals on TAD boundaries, within TADs, or on looping enhancers and promoters. Normalized bigwig files were used as input for calculation by "computeMatrix reference-point" function in deepTools software[62].

**Hi-C data processing**. HiCExplorer (v.3.4.2)[63,64] was used for the processing of Hi-C data. Only uniquely mapped read pairs were saved for further trimming of dangling end reads, the same fragment reads, self-circled reads, self-ligated reads, and other invalid Hi-C reads. Non-duplicated reads were used to generate Hi-C matrix at 5, 10, and 40 kb resolution with cooler (v.0.8.5)[65]. After filtering bins with low counts, the matrices were balanced by Knight–Ruiz (KR) with hicCorrect-Matrix in HiCExplorer. Replicates of Hi-C matrices were normalized to the smallest read count by hicNormalize in HiCExplorer with "–normalize smallest" parameter.

**Visualization of chromatin interaction**. Contact heatmaps were generated with matrices at 5 kb resolution by hicPlotMatrix in HiCExplorer. Curves showing count distribution with genomic distances were performed by hicPlotDistVsCounts in HiCExplorer. PyGenomeTracks (v.1.0) was used to perform and visualize loops and interactions at different regions. Aggregate peak analysis was processed with hicAggregateContacts was used with parameter "–transform obs/exp", which generated aggregate heatmaps and average contact signals.

**Compartment identification**. Compartments were predicted with matrices at 100 kb resolution and regarded as A or B according to the gene density. The compartments with higher gene density were selected as type A, while the compartments with lower gene density were selected as type B[21]. Besides, we judge the transcription activity based on pubic scRNA-seq data of the same cell populations identified by surface markers. That facilitates judgment of Type A or B compartments.

**TAD identification and quantification**. TADs were identified with hicFindTADs at 40 kb resolution. TAD insulation scores (or called TAD separation scores) were generated when identifying TAD boundaries with hicFindTADs tool of HiCExplorer software. According to hicFindTADs tool of HiCExplorer software, an insulation score is used to identify the degree of separation between the left and right regions of each Hi-C matrix bin. The insulation score is measured using the $z$-score of the Hi-C matrix and is defined as the mean $z$-score of all the matrix contacts between the left and right regions[63,64]. The minus value of the insulation score was boundary strength. We calculated the standard deviations of the strength of each TAD boundary across four-cell stages and sorted boundaries by standard deviations. Then the top 1000 variable TAD boundaries were selected based on the ranking order.

Domain scores were calculated to reflect the connectivity within TADs[66,67]. In brief, a domain score was defined as a ratio of the number of intra-TAD paired-end tags to that of all TAD paired-end tags. Domain scores were further normalized by dividing against average domain scores from 1000 simulated TADs with the same size randomly distributed in the same chromosome. Next, the scores were further normalized by subtracting the mean score of all TADs and were treated for quantile-normalization to facilitate comparisons among all Hi-C libraries. The top 1000 variable TADs were chosen by ranking variances of domain scores across four stages, from 2068 significantly changing TADs identified by ANOVA test (with the cutoff FDR < 0.001).

**Loop and interaction identification**. Loops and interactions were detected with hicDetectLoops at 10 kb resolution. To quantify interaction counts between any two lists of genomic regions (loop strength), hicAggregateContacts was used with parameter "–transform obs/exp".

**Classification of loops and interactions**. Loops involving enhancers or promoters were those loops with at least one anchor overlapped with enhancer regions (distal H3K27ac ChIP peaks) or promoter regions (TSS ± 5 kb). Enhancer–promoter (E–P), promoter–promoter (P–P), and enhancer–enhancer (E–E) interactions were classified by overlapping anchor regions with enhancers and promoters. RUNX1-engaged E–P interactions were those with RUNX1 occupancy on anchor regions. RUNX1-engaged E–P interactions were further separated into three types, with RUNX1 occupancy on promoters, enhancers, or both.

**scRNA-seq data analysis**. The scRNA-seq data is existing data, which have been deposited in the NCBI Gene Expression Omnibus under accession number

GSE139389 and GSE67120. The public data of four stages were analyzed together after normalization by log2(TPM/10 + 1).

**Genomic annotation analysis**. Pie chart showing genomic annotations for function regions were performed with R package ChIPseeker (v.1.20.0) and cluster-Profiler (v.3.12.0). We set TSS ± 5 kb regions as the promoter regions. Genes near to peaks or interacting regions were annotated with annotatePeaks.pl in Homer. Only genes whose distances to anchor regions were ≤20 kb were treated as genes influenced by this loop or interaction.

**TF motif analysis**. De novo motifs and known motifs were identified with enhancers (distal H3K27ac ChIP peaks) and promoters (TSS ± 5 kb regions) by findMotifsGenome.pl in Homer. The parameters were set as "-mask -size given -len 8,10,12". Only those motifs whose $P$-values were smaller than 10e−2 were treated as significantly enriched motifs.

**Quantification and statistical analysis**. All statistical analysis was conducted with R version 3.5.1. ANOVA and two-sample Wilcoxon Rank Sum test was used for comparisons of domain scores, histone modification signals, and gene expressions between two adjacent cell populations or replicates. $P$-value < 0.05 was referred as a statistically significant value (if not specified for stricter 0.01 or 0.001). We applied Benjamini–Hochberg correction to correct $P$-values in multiple-testing procedures. FDR < 0.05 after correction was referred as a statistically significant value (if not specified for stricter 0.01 or 0.001). Gene Ontology analysis about the biological process was performed using Metascape (http://metascape.org)[68,69] and GREAT (http://great.stanford.edu/public/html/)[70], with Benjamini–Hochberg $P$-value correction.

**Reporting summary**. Further information on research design is available in the Nature Research Reporting Summary linked to this article.

## Data availability
The data that support this study are available from the corresponding authors upon reasonable request. Hi-C and itChIP-seq data generated in the course of this study have been deposited at the NCBI Gene Expression Omnibus (GEO) with accession number GSE161328. The scRNA-seq data is existing data available in GEO under accession numbers GSE139389 and GSE67120. Bulk RUNX1 ChIP-seq data with other cells for comparison in this study were GSE128767, GSM2224442, GSM2224447 from GEO and SRR5516292, SRR7826002, SRR611749, SRR546127, SRR054918 from The European Bioinformatics Institute (EMBL-EBI). Source data are provided with this paper.

## Code availability
Custom code used in this study is available from the corresponding authors upon reasonable request.

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

## Acknowledgements

We thank Wei Xie Lab at Tsinghua University for sharing the protocol of low-input Hi-C method. We thank all members of the He lab for critical comments on this manuscript. The study was supported by the Peking-Tsinghua Center for Life Sciences (by A.H.), the 1000 Youth Talents Program of China (by A.H.), and the grants from the National Key R&D Program of China (2021YFA1100100, 2017YFA0103400, 2019YFA0801802,

2020YFA0112400, and 2016YFA0100601) (by B.L. and A.H.), the National Natural Science Foundation of China (32025015, 81890991, 31871173, 31930054, 31571487, and 31771607) (by B.L. and A.H.), the Program for Guangdong Introducing Innovative and Entrepreneurial Teams (2017ZT07S347) (by Y. Lan), and the Key Research and Development Program of Guangdong Province (2019B020234002) (by Y. Lan).

## Author contributions

A.H., B.L., and Y. Lan conceived and designed the study. C.L. and G.Z. designed and performed experiments with the help of Y.N., J.Z., Y. Li, S.H., and X.Z. C.L. performed the computational analyses with the help of J.D., D.L., and Z.L. C.L. and A.H. wrote the paper with input from all other authors. All participated in data discussion and interpretation.

## Competing interests

The authors declare no competing interests.
