## [Peer Review File · Nature Communications]

REVIEWER COMMENTS

Reviewer #1 (Remarks to the Author):

The manuscript by Li et al. aims to obtain a better understanding of the transcriptional and epigenetic changes accompanying the generation of the long-term HSCs in the mouse embryo. To this end they have generated Hi-C, ChIP-seq, and RNA-seq datasets on small cell populations of endothelial, hemogenic endothelial (HE) and hematopoietic cells isolated from the E10-E11 AGM and E14.5 fetal liver (Figure 1a,b). This is the first study to report on chromatin conformation in primary cells undergoing EHT, and their datasets will be of interest to the hematopoietic community. They explored histone modifications, Runx1 TF binding, and chromatin conformation changes associated with changes in gene expression over hematopoietic development. From their analyses the authors conclude that regulatory elements active in HSCs are already pre-configured with active histone modifications in arterial endothelial cells, prior to the formation of small scale chromatin loops. In addition, they identified developmental stage specific transcription factors that cooperate with Runx1 during HSC development. While the paper will be of interest to the wider developmental hematopoiesis community, I have some questions/comments regarding the interpretation of the results.

Main points

1. Based on scRNA-seq analyses, the E10 AGM arterial endothelium, termed 'early AEC' in the paper, is presented as the precursor of the HE and HSCs. However, scRNA-seq trajectories do not equal lineage tracing. Indeed, it is well established by the Medvinsky group that the HSC lineage has already diverged from the endothelium at the pro-HSC stage, which cells are generated as early as E9.5 and are the precursors of (pre-)HSCs (Rybtsov et al., Stem Cell Reports 2014). Experiments from the Speck lab in which Runx1 was ectopically expressed in non-hemogenic endothelial cells at different developmental stages also question whether the E10 non-hemogenic endothelium still gives rise to HE and contributes to the HSC population (Yzaguirre et al., Development 2018). This affects part of the interpretation of the data: the clear differences observed between E10 AECs and the E10/E11 EHT populations may not reflect an important transition in EHT, but simply reflect the fact that the E10 AEC is on a different fate trajectory from the E10 HE/pre-HSCs. This should be acknowledged and corrected, for example on page 7 in the description of suppl Fig 2; page 8 "Our genome-wide analysis presented a significantly up-regulated short-range intra-TADs contacts from early AECs to HECs but not in the late developmental populations"; page 14 "...our results indicated that...initiated at early AECs become stronger..."; page 16 "...we followed the continuum of HSC formation starting from ... early AECs..."; page 17 "...pre-established in early AEC and later enhanced in LT-HSCs"; etc.. Finally, given that E10 is midgestation, the term 'early' AEC does not correctly reflect the biology and should be changed to AEC.
2. Page 5, last sentences: "genes on two main boundary clusters" Are genes on the boundary or within the TAD meant here? Also, can the authors clarify how changes to the TAD boundary facilitates the metabolism in hematopoiesis if the metabolism-associated genes there do not change their expression?
3. The detection of Runx1 ChIP peaks in AECs is unexpected. An alternative, and in my view more plausible, explanation would be that the sorting strategy does not completely separate HE from AECs. Particularly since the most defining characteristic of HE is its Runx1 expression. Yet another explanation is that the peaks are due to background in the ChIP. A claim that AECs have Runx1 ChIP-seq peaks would need to be substantiated by functional assays.
4. Page 16: The statement at the end of the first paragraph requires experimental support, e.g. by perturbing Runx1 or co-TF expression and evaluating the interactions, or could to be toned down a little.
5. I missed RNA-seq in the methods and in the statement on data availability. Can the authors please clarify. If this is new data, this should be made available to the scientific community along with the paper. If it is existing data, can this please be clarified.

Minor points

1. In the abstract 'lineage priming' is mentioned. It is not clear what is meant in this context. EHT is known to be accompanied by a replacement of an endothelial programme with a hematopoietic one. This is not what is generally understood under lineage priming.
2. Page 4: "we iteratively quantified..." Can the authors please clarify this sentence.
3. Page 7, middle of page: " C1 with decreased TAD connectivity was accompanied with H3K27ac increase: the figure seems to show a decrease?"
4. Page 16: "Our study provides the first functional annotation..." What is meant by functional?
5. Materials and methods: The activity of type I collagenase is not stopped by FBS, but it is removed by washes.

Reviewer #2 (Remarks to the Author):

In this study, the authors analyzed epigenome state at different stage during early mouse HSCs development, including aAECs, HECs, pre-HSCs and LT-HSCs. Combing the data from 3D genome, histone modification, transcriptome as well as key transcription factor such as RUNX1 occupancy, they generate epigenome dynamics during mouse early HSC development. This study also highlights the role of RUNX1 in regulating the chromatin structure in this process. Overall, the manuscript is well written and provides important data and information in understanding cell lineage decisions from aAECs to HSCs, thus justifies for NC.

Comments:

1. The authors claimed that the active histone modifications on blood related genes were pre-configured in the early stages of cells during early HSC development. While it was true based on the data showing in Fig. 2. However, it seems that the suppressive histone marker, H3K27me3 decreased along the differentiation on these genes (Fig.2b), suggesting that a de-repression also occurred in this process. Please discuss and also revised the final model in Fig.5e.
- 2, In Figure 2a-b, the authors claimed that C1 cluster shows the decreased TAD connectivity accompanied with H3K27ac increase and H3K27me3 reduction. But, it seems that the C1 cluster contains two clusters with distinct patterns, please clarify.
- 3, How the data in Fig.5d was generated needs more detailed description.

Reviewer #3 (Remarks to the Author):

In "Pre-configuring chromatin architecture with histone modifications guides hematopoietic stem cell formation in mouse embryos", Li, Zhang et al. use a combination of itChIP-seq and sisHi-C to probe the changes to chromatin architecture of developing mice as cells undergo the endothelial-to-hematopoietic transition. This work presents a valuable resource to those in the field of chromosome structure and dynamics trying to understand the links between 3D chromosome architecture and gene expression. Moreover, this work provides new in-vivo data relevant to researchers studying the EHT pathway. These experiments are challenging – especially having been done with mouse tissues - given the paucity of materials, and the authors have done a really good job in obtaining high quality Hi-C and ChIP-seq data. This system seems to be of high biological importance and thus the topic matter would be of interest to the general audience of Nature Communications. However, improvements are necessary before it is ready for publication to help the article reach a broader audience. Moreover, a few minor inconsistencies in the figures and text need to be addressed. Finally, a couple of analyses and conclusions have to be revisited or revised. Overall, I believe this article is close to publishable at Nature Communications, and I recommend this article continue undergoing revisions.

General comments:

- Several items were left poorly introduced in the paper, for example:

1. No mention or introduction is made about the protein RUNX1 until late into the paper although it plays a prominent role in the abstract. The reader is left wondering: Why is it important? Why is it used as a marker? I suggest that the authors introduce RUNX1, its general importance and why they chose it as a marker for the EHT.
2. itChIP-seq was not well defined beyond a passing mention. How it differs from the canonical ChIP-seq methodology is left to the reader to dig into the literature. I strongly suggest that the authors briefly clarify the methodological differences between itChIP-seq and ChIP-seq, and clearly define what itChIP-seq stands for.

In the abstract the sentence: "RUNX1 and co-TFs together constituted a central, progressively intensified enhancer-promoter interaction hub" does not seem to be strongly supported by the data and analysis (see my concerns below in the point-by-point comments). I strongly suggest that the authors revise this conclusion (and any results related to Fig. 4f). The analysis leading to the definition of the "RUNX1 interaction hub" appears highly qualitative, and left this reviewer wondering – how robust is the analysis? Does it add any understanding of the EHT system? In my opinion, this aspect of the paper can use a significant revision, and could be almost entirely removed to improve the quality of the paper.

Point-by-point comments/concerns (in roughly chronological order):

The authors state: "In general, enhancers are known to control the gene transcriptional activity and specificity via looping to target promoters, which are predicted to most likely occur within TADs." I believe this statement is factually incorrect. Other mechanisms of E-P interaction have also been proposed which do not involve direct E-P looping. In general, the mechanisms of E-P interactions are poorly understood (see for example: Brandao et al. Current Opinion in Cell Biology 70 (2021): 18-26.)

Fig1: In panel (g) the authors say that gene expression profiles are calculated as: $\log_2(\text{TPM}/10+1)$. However, values below 0 can be seen from the violin plots, which should not be possible as the minimal value of TPM is 0, and $\log_2(1)=0$. The authors should review why negative values are appearing in their violin plots.

I thank the authors for including summary statistics for their low input Hi-C and itChIP-seq (Supplementary Table 1). I am, however, slightly concerned about inconsistency of reporting of the values - specifically about the input cell numbers. In some rows, the values are nicely rounded (1000, 1000, 1000, 500, etc.) In other rows, the values are highly specific (286, 359, 911). How was cell counting done? Are these values estimates? Are they measurements? What are the error bars? Please specify.

In the methods, the authors write: "DNA was purified and extracted by pheno-chloroform". I think this is a typo, and they meant "phenol-chloroform".

In the methods, the authors write "Domain scores were calculated as mentioned before [Ref. 57]". Please provide the score calculation directly in the methods or supplementary text.

In the quantification and statistical analysis section of the methods, the authors say that $P < 0.05$ was considered significant unless otherwise specified – the authors should consider correcting the p-values for the number of tests performed (e.g. Bonferroni correction, etc).

In supplementary Figure 3, the figure legends for the aggregate peak analysis says the window is a 90

kb x 90 kb. However, the axis labels indicate - 75 kb to 75 kb, and the window is 19 by 19 pixels (suggesting the window is ~190 kb by 190 kb). Please correct the discrepancy.

In supplementary Figure 2 g, it is difficult to judge the sizes of the TADs, loops and compartments solely based on the gene annotations and the written value: chr1:69,315,710-73,687,649. I suggest that the authors provide a scale bar, or some measure of the size of the window or the pixel sizes.

Figure 1c legend is misleading. The legend states that Type A compartments were defined by actively transcribing genes, while Type B compartments by silenced genes. However, at this stage the authors have not performed any RNA-seq, and it appears in the methods that the compartments are actually defined by Hi-C maps binned at 100 kb. Perhaps the authors mean that the Type A and Type B regions were identified by Hi-C and Type A was selected as the region with higher gene density (as in their methods). I suggest that the author revise this sentence.

In Figure 1d, the y-axis label is missing. What are the units? What are the values? Within a 10 kb bin, how are values aggregated – is it sum, mean, median?

Figure 1e, what are the units on the colour bars? Why is the insulation score both positive and negative – how was it calculated (the methods only refer the reader to another paper...)?

Figure 1g: I am confused about how a single p-value is reported for all of the comparisons between gene-expression between the violin plots. Also, the two sentences reporting the p-values seem incomplete: e.g. "Genes in the increasing-strength TAD boundaries, Kruskal-Wallis, $p = 0.0011$ " is lacking a verb to indicate where the reader attention should be focused. Can the authors please clarify this?

The authors claim that "... H3K27me3... signals were markedly accumulated within TADs". However, the data does not appear to support this claim. In Figure 1d, the profile for H3K27me3 is rather flat within the body of the TAD (i.e. unlike the other profiles, there is no discernible shape). Instead, the data could instead suggest that H3K27me3 is depleted at the TAD boundary more than accumulating within the TAD. The last point to note regarding H3K27me3 is that is that the itChIP-seq signals are the least reproducible (e.g. showing only ~60-70% pearson correlations between replicates, and sometimes higher correlations with other cell types). Due to the poor reproducibility of H3K27me3 signals with itself, I suggest that the authors be more cautious in their statement, or revise it to be more consistent with their data.

For Figures 1e-g why did the authors choose the top 1000 variable TADs? How were these variable TADs defined? How would the results change if the authors had chosen the top 2000, or 3000 TADs, or only the top 100 TADs? How robust are these results? Can the same conclusions be drawn if instead all TADs are used as in Supplementary Figure 1g?

Related to Figure 1e, the main text says: "This indicated a role implicated in HSC dormancy, as reported elsewhere". What element is playing the role (is the role for H3K4me3, or boundary strength, or something else)?

The author's conclusion remark: "these data support that the TAD boundary dynamics is dramatic, facilitating the metabolism activity for hematopoiesis" is not supported by the data. The authors have only shown a partial correlation between changes in gene expression (e.g. from a seemingly selectively chosen GO-term expression analysis) and changes to boundary strengths. Indeed, most cells do not show any significant sign of change in gene expression – e.g. even in the author's own data it only occurs between LT versus the other cell types. I strongly suggest the authors revise this statement or provide more direct evidence (some suggestions below).

Suggestions to the point above: 1a) To show what fraction of TAD boundaries undergo significant changes: how much of the change is due to noise in the data versus intrinsic to the biological system? The authors could test this, for instance, by downsampling or re-sampling the data, and comparing across conditions or within conditions. 1b) The authors used the insulation score to define TAD boundaries, which does not distinguish between compartments and TADs – can the authors differentiate between contributions due to A/B compartments and TAD boundaries? If so, how did they do it? 2) In the GO-term analysis what p-value is even deemed significant when multiple tests are being performed (i.e. what is the Bonferroni – or other - corrected p-value?) 3) As far as the claim that TAD dynamics “facilitate the metabolism activity for hematopoiesis”, if this statement is to be made, perhaps the authors could look at other cell types in data available elsewhere – is there any evidence in any cell type that TAD insulation is strong in these regions in the absence (or presence) of gene expression?

In Supplementary Figure 2a, are the different columns representing technical or biological replicates? Please specify.

For Figs. 2a-b again, why are only 1000 TADs chosen, how do the results change with other numbers of TADs? What are the corrected p-values for multiple hypothesis testing significance for the GO-term analysis?

The statement “C1 with decreased TAD connectivity was accompanied with H3K27ac increase and H2K27me3 reduction” is seemingly inconsistent with the figure 2 a-b. From the pre to LT columns, H3K27ac goes from more red to more blue suggesting a decrease. Did the authors mean H3K27ac decrease? Does that change the author’s conclusions?

Generally, the comparisons in the paragraph: “C2 with a sharp decrease in TAD connectivity between early AECs and HECs displayed higher H3K27me3 and lower H3K4me1 and nearly complete depletion of H3K27ac and H3K4me3. C3 with a gradually increasing TAD connectivity showed strong H3K27ac, H3K4me1 and H3K4me3 signals as early as eAECs with a slow reduction in H3K27me3. Interestingly, genes within C3 were associated with several hematopoiesis related functional terms...” needs quantification. How significant are the differences that the authors highlight? More importantly, what are the fold-changes in the signal values?

“Specific inspection of TAD scores and histone modifications at the Runx1 exemplified the increased intra-TAD contacts. This identified the newly engaged P1 promoter in HECs evidenced by H3K4me3 signals and the enhancer activation by H3K27ac signals (Fig.2e).” It appears that much of what the authors are showing in the insets (i.e. higher Hi-C signal) can be attributed to the change in the contact probability decay genome-wide that occurs between eAECs and the later cell types (see Fig. 2f). Can the authors compute the observed over expected Hi-C scores and show that specific interactions appear after removing the contribution due to the changing overall contact probability scaling with genomic distance? Perhaps deeper sample sequencing may be required to reveal a specific looping-type interaction. Can the authors quantify (similarly to Supl. Fig. 3c) the change in the loop strength at RUNX1 – what is the effect size/ fold-change over time?

Figure 3a – how did authors identify loops – were all the data pooled together? The authors used a loop calling algorithm, but loop calling is generally very difficult – especially with noisy data with only a few million reads. Can the authors show specific examples of good and bad loops called by their data? What fraction of loops called by their algorithm usage were nicely visible in their Hi-C maps?

In the figure 3a-3b colour maps, what are the colour bar units, and how were the values calculated/what do they represent, e.g. $\log_2(\text{value})$, $\log_{10}(\text{value})$, $\text{value}/\text{input}$?

In Figure 3e, how are the “interactions” on top of the tracks defined? The authors refer to a software package in the methods that generate them, but what do the results mean? Are the shown interactions (or changes to interactions) robust? Was Hi-C used to generate the arches (shown on top of the tracks), or was the track view generated only from the itChIP-seq tracks directly?

In Figure 3f, why is the violin plot showing negative values of gene expression?

The authors establish a “chromatin looping hub” for RUNX1 using Cytoscape. Can the authors explain how this graphical network helps establish that “core chromatin interactions among some hematopoiesis genes initiated at early AECs become stronger were likely mediated by steadily increasing RUNX1 expression and binding signals at later developmental stages”? The analysis seems tautological – the authors first identify a hub via RUNX1, then label interactions within the hub, then look at changing interactions within a hub, seeing that some become stronger within a specific subset within the hub, and use this to claim that the increasing strength of interaction is “likely mediated by steadily increasing RUNX1 expression”. I strongly suggest this analysis to be removed and the conclusions revisited, unless it can be appropriately explained, independently verified and quantitatively supported.

Why define a looping hub only with C2 genes when RUNX1/H3K27ac signals are also well correlated in the other clusters?

In Figure 5a, how is “distal” defined? What is the genomic distance?

In Fig. 5, the authors use the term “RUNX1-mediated interactions”. How can the authors make this claim? What is the evidence supporting that RUNX1 is casual (i.e. mediating) the interactions?

Reviewer #1 (Remarks to the Author):

The manuscript by Li et al. aims to obtain a better understanding of the transcriptional and epigenetic changes accompanying the generation of the long-term HSCs in the mouse embryo. To this end they have generated Hi-C, itChIP-seq, and RNA-seq datasets on small cell populations of endothelial, hemogenic endothelial (HE) and hematopoietic cells isolated from the E10-E11 AGM and E14.5 fetal liver (Figure 1a,b). This is the first study to report on chromatin conformation in primary cells undergoing EHT, and their datasets will be of interest to the hematopoietic community. They explored histone modifications, Runx1 TF binding, and chromatin conformation changes associated with changes in gene expression over hematopoietic development. From their analyses the authors conclude that regulatory elements active in HSCs are already pre-configured with active histone modifications in arterial endothelial cells, prior to the formation of small-scale chromatin loops. In addition, they identified developmental stage specific transcription factors that cooperate with Runx1 during HSC development. While the paper will be of interest to the wider developmental hematopoiesis community, I have some questions/comments regarding the interpretation of the results.

Main points

1. Based on scRNA-seq analyses, the E10 AGM arterial endothelium, termed 'early AEC' in the paper, is presented as the precursor of the HE and HSCs. However, scRNA-seq trajectories do not equal lineage tracing. Indeed, it is well established by the Medvinsky group that the HSC lineage has already diverged from the endothelium at the pro-HSC stage, which cells are generated as early as E9.5 and are the precursors of (pre-)HSCs (Rybtsov et al., Stem Cell Reports 2014). Experiments from the Speck lab in which Runx1 was ectopically expressed in non-hemogenic endothelial cells at different developmental stages also question whether the E10 non-hemogenic endothelium still gives rise to HE and contributes to the HSC population (Yzaguirre et al., Development 2018). This affects part of the interpretation of the data: the clear differences observed between E10 AECs and the E10/E11 EHT populations may not reflect an important transition in EHT, but simply reflect the fact that the E10 AEC is on a different fate trajectory from the E10 HE/pre-HSCs. This should be acknowledged and corrected, for example on page 7 in the description of suppl Fig 2; page 8 "Our genome-wide analysis presented a significantly up-regulated short-range intra-TADs contacts from early AECs to HECs but not in the late developmental populations"; page 14 "...our results indicated that...initiated at early AECs become stronger..."; page 16 "...we followed the continuum of HSC formation starting from ... early AECs..."; page 17 "...pre-established in early AEC and later enhanced in LT-HSCs"; etc.. Finally, given that E10 is midgestation, the term 'early' AEC does not correctly reflect the biology and should be changed to AEC.

Reply: We thank the reviewer for pointing out this important issue. Overall, we agree that scRNA-seq trajectories do not equal lineage tracing, while the evidence of precise lineage tracing is still lacking to decisively conclude when the hemogenic fate of AECs in dorsal aorta is fully resolved. However, the aforementioned studies by both Medvinsky

and Speck labs do not exclude the possibility for at least part of E10 AECs as origins of HSC-priming HECs. More recently, studies from the Kai Tan lab¹ and the Lacaud lab² suggest that the transition of AECs to HECs in dorsal aorta still occurs at E10.5. The functional evidence from the Lacaud lab further supports that AECs from E10.5 dorsal aorta marked by ACE expression have hemogenic potential. These findings strongly argue that the fate of AECs is not yet fully resolved immediately after E10. Supportively, a relevant genetic lineage tracing study using Cx40-CreER has also evidenced the contribution of arterial ECs (marked by Cx40 expression) to hematopoietic cells in liver and thymus at E12.5 when induced labeling at E9.5³. Considering that the Cre activity *in vivo* takes effect several hours after tamoxifen injection, the transition of AECs to HECs should occur around E10. We have previously identified two transcriptomically different AEC populations in mid-gestational dorsal aorta, namely early AECs and late AECs, which were sampled mainly from E9.5-E10.0 and E10.5-E11.0, respectively. This led to the view that early AECs lie upstream of both late AECs and HECs, verified by the previous CFU-C assay⁴. Thus, late AECs rather than early AECs take on a different fate trajectory from E10 HECs⁴. Therefore, these findings serve as the important basis for the present study as well as the reason for the annotation of “early AEC” here. We also believe that future precise physiological lineage tracing is required to determine when the fate of AECs in dorsal aorta is fully resolved.

2. Page 5, last sentences: “genes on two main boundary clusters” Are genes on the boundary or within the TAD meant here?

Reply: “genes on two main boundary clusters” means genes on the boundaries, but not within TADs.

-- Also, can the authors clarify how changes to the TAD boundary facilitates the metabolism in hematopoiesis if the metabolism-associated genes there do not change their expression?

Reply: We appreciate this critical point. Indeed, we did not observe significant changes in transcription associated with dynamic TAD boundaries across four populations. Since metabolic pathways have been established to be involved in hematopoiesis across many stages⁵, we reasoned that no significant changes in transcription likely lie with two possibilities: 1. Transcriptional alteration may take place later beyond this developmental window than these boundary strength changes as detected here; 2. We cannot exclude the current technical limitation with scRNA-seq datasets from each population. Therefore, we modified our description to “Despite of no detected alterations in transcription, examination of the TAD boundary strength dynamics may help predict early changes in metabolic pathways required for hematopoiesis”.

3. The detection of *Runx1* ChIP peaks in AECs is unexpected. An alternative, and in my view more plausible, explanation would be that the sorting strategy does not completely separate HE from AECs. Particularly since the most defining characteristic of HE is its *Runx1* expression. Yet another explanation is that the peaks are due to background in the ChIP.

Reply: We are sorry for the confusion. First, we believe that the discrepancy is likely due to the definition of eAECs (early arterial endothelial cells). The eAECs termed here in contrast to late AECs are not mature AECs, but a specific subpopulation of early progenitors which can differentiate into HECs or mature AECs. Our previous in vitro functional study⁴ showed that eAECs can differentiate into either mature AECs or HECs. To our surprise, we did not separate eAECs and HECs by *RUNX1*, although *Runx1* expression in eAECs is lower than in HECs (Reviewer figure 1a,b).

However, we also agree with the reviewer that the sorting strategy does not guarantee 100% separation of HECs from potential non-HECs. That might lead to the *RUNX1* expression and ChIP peaks. However, except for the shared peaks, specific *RUNX1* peaks were also identified in eAECs and HECs (Reviewer figure 1c), suggesting the bona fide *RUNX1* binding rather than the background from eAECs. In addition, we believe that future technical advances by performing single-cell *RUNX1* ChIP-seq may yield more accurate details.

Reviewer figure 1. FACS strategy and *Runx1* expression in eAECs and HECs.

a Surface markers for FACS for eAECs and HECs. **b** Violin plot showing *Runx1* expression in eAECs and HECs. The normalized expression values were calculated as $\log_2(\text{TPM}/10+1)$ from previously public scRNA-seq data (GSE139389 and GSE67120) of cell populations defined using the same surface markers. Wilcoxon test was used to determine the statistic significance. **c** Heatmap showing the common and specific RUNX1 peaks in four cell populations. The ± 500 bp regions around all peak centers merged from four cell populations were used for calculation.

To further exclude the plausible eAECs “peaks” due to background, we analyzed our RUNX1 datasets with other public RUNX1-ChIP-seq but from other cell types at specific regions (Supplementary Fig. 5a) and genome wide (Supplementary Fig. 5b). This supports that RUNX1 peaks identified in our data are biologically meaningful and highly enriched for the RUNX1 motif (P -value $< 1e-17$ by Homer).

Supplementary Fig. 5: Comparison of RUNX1 itChIP-seq data with public RUNX1 ChIP-seq data of other cell types. **a** Track view showing the visualization comparison of our RUNX1 itChIP-seq data with public RUNX1 ChIP-seq data from other cell types. There were common peaks sharing the same binding motif, as well as unique peaks reflecting the cell type specificity. **b** Heatmaps showing the cell type specific enrichment signals of RUNX1 binding in our data and public data. The regions were peaks merged from all seven RUNX1 ChIP-seq data sets. Regions in the upper part heatmaps were sorted by descent RUNX1 signals around peak-center ± 1 kb in SRR5516292-ChIP. Regions in the lower part heatmaps were sorted by descent RUNX1 signals around peak-center ± 1 kb regions in eAEC RUNX1-itChIP. eAEC, early AEC; pre, pre-HSC; LT, LT-HSC.

-- A claim that AECs have Runx1 ChIP-seq peaks would need to be substantiated by functional assays.

Reply: Indeed, previous studies by immunostaining for RUNX1 in dorsal aorta have shown that RUNX1 is not only expressed in the ventral DA, potent to differentiate into HECs, but also in other ECs of dorsal DA^{4, 6, 7}. This suggests that only a portion of RUNX1⁺ ECs ultimately lead toward HECs. In addition, eAECs display lower expression of RUNX1 as described above.

(Fig.5a in Hou et al., Cell Research, 2020)⁴

4. Page 16: The statement at the end of the first paragraph requires experimental support, e.g. by perturbing Runx1 or co-TF expression and evaluating the interactions, or could to be toned down a little.

Reply: Thanks for this suggestion. In this revision, we tone down the statements as following: It is tempting to speculate that RUNX1 and co-TFs identified within E-P interactions might function as a TF module to license gene transcriptional specificity.

5. I missed RNA-seq in the methods and in the statement on data availability. Can the authors please clarify. If this is new data, this should be made available to the scientific community along with the paper. If it is existing data, can this please be clarified.

Reply: We are sorry for the missing information. The scRNA-seq data are from existing data, deposited in the NCBI Gene Expression Omnibus under accession number GSE139389 and GSE67120. In this revision, we specify this in Methods and in the statement on data availability.

Minor points

1. In the abstract 'lineage priming' is mentioned. It is not clear what is meant in this context. EHT is known to be accompanied by a replacement of an endothelial programme with a hematopoietic one. This is not what is generally understood under lineage priming.

Reply: We change the statement as following. Although definitive hematopoietic stem cells (HSCs) are known to emerge via endothelial-to-hematopoietic transition (EHT), how the multi-layered epigenome is sequentially unfolded in a small portion of endothelial cells (ECs) transitioning into the hematopoietic fate remains elusive.

2. Page 4: "we iteratively quantified..." Can the authors please clarify this sentence.

Reply: We quantified the dynamic trajectories in different scales of the 3D genome structure until the scale was identified to correlate well with dynamic histone modifications. To avoid potential misunderstanding, we remove "iteratively" in this revision.

3. Page 7, middle of page: "C1 with decreased TAD connectivity was accompanied with H3K27ac increase: the figure seems to show a decrease?"

Reply: Thanks for pointing out this typo. It should read "C1 with decreased TAD connectivity was accompanied with H3K27ac decrease".

4. Page 16: "Our study provides the first functional annotation..." What is meant by functional?"

Reply: We mean the biological processes as annotated by the Gene Ontology (GO) term analysis of cis-regulatory regions identified by multi-omics profiling. To be clarified, the

sentence is now modified to: “Our study provides the first biological annotation of the dynamic 3D genome structure in combination with histone modifications, TF binding and transcriptome during early HSC development.”

5. Materials and methods: The activity of type I collagenase is not stopped by FBS, but it is removed by washes.

Reply: Yes, we corrected this description in Materials and Methods: “Primary embryonic single-cell suspension was acquired using the type I collagenase for dissociation at 37°C and PBS/5% FBS for washes.”

Reviewer #2 (Remarks to the Author):

In this study, the authors analyzed epigenome state at different stage during early mouse HSCs development, including eAECs, HECs, pre-HSCs and LT-HSCs. Combining the data from 3D genome, histone modification, transcriptome as well as key transcription factor such as RUNX1 occupancy, they generate epigenome dynamics during mouse early HSC development. This study also highlights the role of RUNX1 in regulating the chromatin structure in this process. Overall, the manuscript is well written and provides important data and information in understanding cell lineage decisions from eAECs to HSCs, thus justifies for NC.

Comments:

1, The authors claimed that the active histone modifications on blood related genes were pre-configured in the early stages of cells during early HSC development. While it was true based on the data showing in Fig. 2. However, it seems that the suppressive histone marker, H3K27me3 decreased along the differentiation on these genes (Fig.2b), suggesting that a de-repression also occurred in this process. Please discuss and also revised the final model in Fig.5e.

Reply: We appreciate the reviewer's recognition of the significance and importance of this work as well as this specific comment. We also agree that a de-repression mechanism is also engaged. Indeed, upon H3K27ac increase in these loci, H3K27me3 decrease is accompanied thereafter. This suggestion is well taken in this revision. We now incorporate this change in the working model (Fig. 5e).

Fig. 5e. Working model of multi-omics dynamics during HSC formation. RUNX1 occupies target promoters or/and enhancers primed with H3K27ac in eAECs. Repressive H3K27me3 gradually diminishes accompanying later H3K27ac increase. Other hematopoiesis related TFs are recruited to form a TF module with RUNX1 to facilitate enhancer-promoter interactions driving the core HSC gene expression.

2, In Figure 2a-b, the authors claimed that C1 cluster shows the decreased TAD connectivity accompanied with H3K27ac increase and H3K27me3 reduction. But, it seems that the C1 cluster contains two clusters with distinct patterns, please clarify.

Reply: We are sorry for the confusion in appreciating C1 cluster in Fig.2a-b. Indeed, C1 cluster refers to the second cluster, while the first cluster now labeled as C0 with only a

small number of TADs likely represents less biologically meaningful information and is therefore removed in the further analysis.

3, How the data in Fig.5d was generated needs more detailed description.

Reply: In this revision, we follow this suggestion by adding more details. Specifically, Fig.5d was generated based on Fig.5b and Fig.5c. First, TF motifs were identified by homer with enhancers and promoters of three classes of RUNX1-engaged E-P interactions in Fig.5b, separately. The corresponding TFs were potentially candidate co-TFs interacting with RUNX1. By clustering with gene expressions of potential RUNX1-interacting TFs (Fig.5c), we focused on those with increased or high expression levels during development (Fig.5c) and presented TF motif significance and expression levels together in Fig.5d.

Reviewer #3 (Remarks to the Author):

In “Pre-configuring chromatin architecture with histone modifications guides hematopoietic stem cell formation in mouse embryos”, Li, Zhang et al. use a combination of itChIP-seq and sisHi-C to probe the changes to chromatin architecture of developing mice as cells undergo the endothelial-to-hematopoietic transition. This work presents a valuable resource to those in the field of chromosome structure and dynamics trying to understand the links between 3D chromosome architecture and gene expression. Moreover, this work provides new in-vivo data relevant to researchers studying the EHT pathway. These experiments are challenging – especially having been done with mouse tissues - given the paucity of materials, and the authors have done a really good job in obtaining high quality Hi-C and ChIP-seq data. This system seems to be of high biological importance and thus the topic matter would be of interest to the general audience of Nature Communications. However, improvements are necessary before it is ready for publication to help the article reach a broader audience. Moreover, a few minor inconsistencies in the figures and text need to be addressed. Finally, a couple of analyses and conclusions have to be revisited or revised. Overall, I believe this article is close to publishable at Nature Communications, and I recommend this article continue undergoing revisions.

Reply: We greatly appreciate that the reviewer recognized the data value and high biological importance as well as the opportunity for the revision.

General comments:

- Several items were left poorly introduced in the paper, for example:

1. No mention or introduction is made about the protein RUNX1 until late into the paper although it plays a prominent role in the abstract. The reader is left wondering: Why is it important? Why is it used as a marker? I suggest that the authors introduce RUNX1, its general importance and why they chose it as a marker for the EHT.

Reply: This suggestion is well taken in this revision. We add them in the Introduction in this revision.

“RUNX1 was a master TF implicated in many stages of hematopoiesis, including EHT in early embryos as well as in leukemia development^{8,9}. The germline deletion of Runx1 resulted in the LT-HSC elimination and the embryonic lethality by E12.5^{10,11}. RUNX1 together with other hematopoiesis-related TFs, GFI1, SPI1 and FOSB (FGRS), has been shown to *in vitro* reprogram endothelial cells into HSCs¹². Several studies have further characterized RUNX1 binding target genes in cell cultures or bulk *in vivo* samples of adults^{13,14,15}. However, thus far, none of them has examined how RUNX1 regulates gene transcription in the context of defined chromatin states in embryos *in vivo*. Further, the integrated knowledge of epigenomic configurations of the 3D genome structure, histone modifications and transcription factors in transcriptional regulation to promote EHT is still lacking.”

2. itChIP-seq was not well defined beyond a passing mention. How it differs from the canonical ChIP-seq methodology is left to the reader to dig into the literature. I strongly suggest that the authors briefly clarify the methodological differences between itChIP-seq and ChIP-seq, and clearly define what itChIP-seq stands for.

Reply: We thank the reviewer for this important suggestion to help improve this manuscript. In this revision, we clearly define the itChIP-seq in the text:

“Different from tedious steps involving the sonication and ligation in traditional ChIP-seq, itChIP-seq adopts Tn5-transposase-based tagmentation as the strategy for simultaneous chromatin fragmentation and barcoded adaptor ligation for the one-step PCR enrichment. itChIP-seq allows for highly sensitive profiling of histone modifications and TFs in low input samples as few as 100 cells³⁰.”

In the abstract the sentence: “RUNX1 and co-TFs together constituted a central, progressively intensified enhancer-promoter interaction hub” does not seem to be strongly supported by the data and analysis (see my concerns below in the point-by-point comments). I strongly suggest that the authors revise this conclusion (and any results related to Fig. 4f). The analysis leading to the definition of the “RUNX1 interaction hub” appears highly qualitative, and left this reviewer wondering – how robust is the analysis? Does it add any understanding of the EHT system? In my opinion, this aspect of the paper can use a significant revision, and could be almost entirely removed to improve the quality of the paper.

Reply: We thank the reviewer for this constructive suggestion. We also agree with the reviewer that this analysis on the RUNX1 interaction hub appears qualitative. Thus, we remove the previous Fig. 4f. Instead, we add sub-clustering and quantification analysis to explore how RUNX1-engaged E-P interactions among C2 genes change in Fig. 4. The new analysis in Fig. 4f,g shows that RUNX1-engaged E-P interaction strengths were enhanced within the gene loci linked to definitive hematopoiesis while decreased within other loci related to mitotic cell cycle and angiogenesis during development.

“Sub-clustering by the interaction strengths showed that RUNX1-engaged E-P interactions with decreased strengths were linked to mitotic cell cycle and angiogenesis (Fig. 4f and Supplementary Fig. 6a). The interaction strengths within the gene loci enriched for the GO terms, B cell proliferation and interferon α/β signaling, appeared transiently higher in HECs and pre-HSCs (Fig. 4f and Supplementary Fig. 6a). Interestingly, the increasing sub-cluster was related to T cell activation, adaptive immune and definitive hemopoiesis (Fig. 4f and Supplementary Fig. 6a). Statistical analyses were further performed in Fig. 4g.”

Fig. 4f**Fig. 4g**
Fig. 4f Lines reflecting strengths of three sub-clusters of RUNX1-engaged E-P interactions among C2 genes. Those RUNX1-engaged E-P interactions existing from eAEC to LT-HSC were used for clustering. The black lines were fitted for the interaction dynamics in each cluster. The thick white bars represented a 95% confidence interval.

Fig. 4g Violin plots quantifying the strengths of three sub-clusters of RUNX1-engaged E-P interactions among C2 genes, corresponding to (f). Box-and-whiskers plots represented the maxima, 75th percentile, median, 25th percentile, and minima. The two-sample Wilcoxon Rank Sum test was performed between two adjacent stages. *, P -value <0.05; **, P -value <0.01; ***, P -value <0.001; ****, P -value <0.0001; "ns", not significant.

Supplementary Fig. 6a

Supplementary Fig. 6a Biological processes in Gene Ontology analysis with genes in three sub-clusters of C2 RUNX1-engaged E-P interactions in Fig.4f.

Therefore, we also make modifications to the related conclusion as following:

“Taken together, our results indicated that RUNX1-engaged E-P interactions were highly engaged in hematopoiesis as early as in eAECs, which were reinforced within EHT.”

Point-by-point comments/concerns (in roughly chronological order):

The authors state: “In general, enhancers are known to control the gene transcriptional activity and specificity via looping to target promoters, which are predicted to most likely occur within TADs.” I believe this statement is factually incorrect. Other mechanisms of E-P interaction have also been proposed which do not involve direct E-P looping. In

general, the mechanisms of E-P interactions are poorly understood (see for example: Brandao et al. *Current Opinion in Cell Biology* 70 (2021): 18-26.)

Reply: We agree that in addition to the direct E-P looping for E-P interactions, other mechanisms have also been revealed. Hereby, we change this description to “direct E-P looping is one of central mechanisms, by which enhancers target promoters and control the gene transcriptional activity and specificity. These interactions are predicted to likely occur within TADs.”

Fig1: In panel (g) the authors say that gene expression profiles are calculated as: $\log_2(\text{TPM}/10+1)$. However, values below 0 can be seen from the violin plots, which should not be possible as the minimal value of TPM is 0, and $\log_2(1)=0$. The authors should review why negative values are appearing in their violin plots.

Reply: Following this comment, we fully examine the calculation, finding that the “negative values” shown in figures are due to the confusing graphic presentations generated by ggpubr package, with ‘trim’ function set as ‘FALSE’.

First, we checked the $\log_2(\text{TPM}/10+1)$, confirming that there are no negative values. The matrices of $\log_2(\text{TPM}/10+1)$ are also attached. Second, we looked back to the source codes of the ggpubr package. The ggpubr package utilizes the ggplot package. The parameter “trim” is TRUE by default in the ggplot package. However, the parameter “trim” is FALSE by default in the ggpubr package. If TRUE, “trim” function will trim the tails of the violins to the range of the data. If FALSE, it does not trim the tails. The tails are generated by smoothing during fitting the distribution. For clarification, we set the parameter “trim” to TRUE.

When we present every point in jitter plot and violin plot together, no negative values are observed as well (Reviewer figure 2).

Reviewer figure 2. Comparison of violin plots generated by R packages with ‘trim’ function as ‘FALSE’ or ‘TRUE’. The violin plots on the right are now shown in Fig.1g. We also revise Fig.1g to include pairwise comparisons between any two populations through two-sample Wilcoxon Rank Sum test.

I thank the authors for including summary statistics for their low input Hi-C and itChIP-seq (Supplementary Table 1). I am, however, slightly concerned about inconsistency of reporting of the values - specifically about the input cell numbers. In some rows, the values are nicely rounded (1000, 1000, 1000, 500, etc.) In other rows, the values are highly specific (286, 359, 911). How was cell counting done? Are these values estimates? Are they measurements? What are the error bars? Please specify.

Reply: We thank the reviewer to pointing out this issue. Indeed, cells in specific numbers were counted by FACS. When cells in one tube obtained by FACS were all used for one experiment, the numbers are likely specific (e.g. 286, 359, 911). When cells in one tube by FACS were aliquoted for separate ChIP experiments, a rounded number of cells were used. For example, when there were 5,132 cells in one tube by FACS from 10 embryo AGMs, we divided them into 5 groups for ChIP experiments. The estimated cell number of each ChIP was $5,132 / 5 \approx 1,000$ cells. Hence, there is no error bars involved for this kind of estimation.

In the methods, the authors write: "DNA was purified and extracted by pheno-chloroform". I think this is a typo, and they meant "phenol-chloroform".

Reply: Thanks for this correction. It should be "phenol-chloroform".

In the methods, the authors write "Domain scores were calculated as mentioned before [Ref. 57]". Please provide the score calculation directly in the methods or supplementary text.

Reply: Thanks for the suggestion. We add the details of the domain score calculation in the Methods in this revision.

"Domain scores were calculated as previously described^{16, 17}. In brief, a domain score was defined as a ratio of the number of intra-TAD paired-end tags to that of all TAD paired-end tags. Domain scores were further normalized by dividing against average domain score from 1,000 simulated TADs with the same size randomly distributed in the same chromosome. Next, the scores were further normalized by subtracting the mean score of all TADs, and were treated for quantile-normalization to facilitate comparisons among all Hi-C libraries."

In the quantification and statistical analysis section of the methods, the authors say that $P < 0.05$ was considered significant unless otherwise specified – the authors should consider correcting the p-values for the number of tests performed (e.g. Bonferroni correction, etc).

Reply: We appreciate this important suggestion. Thus, we followed this to include p-value correction whenever applicable. However, we choose Benjamini-Hochberg correction rather than Bonferroni correction. When the n is large (such as 4839 TADs in my case), the Bonferroni correction will lose many varying candidates and potential biological meanings. Therefore, we re-examine the *P*-values with the multiple-testing procedures and include Benjamini-Hochberg correction. Overall, this correction does not change the patterns and conclusions. I also add the statement of the *P*-value correction in this revision.

In supplementary Figure 3, the figure legends for the aggregate peak analysis says the window is a 90 kb x 90 kb. However, the axis labels indicate - 75 kb to 75 kb, and the window is 19 by 19 pixels (suggesting the window is ~190 kb by 190 kb). Please correct the discrepancy.

Reply: We are sorry for this confusion. Each pixel was 10 kb x 10 kb. In x or y axis, the region shown was from -90 kb (-9 pixels) to + 90 kb (+9 pixels) around the center 10-kb pixel. The square was 19 x19 pixels, the same as 190 kb x 190 kb. The previous axis labels were indeed incorrect and should be changed to -90 kb to + 90 kb in this revision.

In supplementary Figure 2 g, it is difficult to judge the sizes of the TADs, loops and compartments solely based on the gene annotations and the written value: chr1:69,315,710-73,687,649. I suggest that the authors provide a scale bar, or some measure of the size of the window or the pixel sizes.

Reply: Following this suggestion, the scale bar and axis labels are added into the figure for better annotations.

Figure 1c legend is misleading. The legend states that Type A compartments were defined by actively transcribing genes, while Type B compartments by silenced genes. However, at this stage the authors have not performed any RNA-seq, and it appears in the methods that the compartments are actually defined by Hi-C maps binned at 100 kb. Perhaps the authors mean that the Type A and Type B regions were identified by Hi-C and Type A was selected as the region with higher gene density (as in their methods). I suggest that the author revise this sentence.

Reply: We greatly appreciate these comments. Indeed, we judge the transcription activity based on pubic scRNA-seq data of the same cell populations identified by surface markers. That facilitates judgement of Type A or B compartments.

In Figure 1d, the y-axis label is missing. What are the units? What are the values? Within a 10 kb bin, how are values aggregated – is it sum, mean, median?

Reply: Thanks for pointing out the missing information. The y units and values are CHIP-seq signals of histone modifications, normalized and aggregated in each 10 kb bin. The aggregated values are mean values. The y-axis labels are added in the revision.

Figure 1e, what are the units on the colour bars?

Reply: From left to right on the bottom of Fig. 1e, the units on the color bars are (1) the values of boundary strengths after row-scaled normalization, (2) gene number within each TAD boundary region, and (3) normalized H3K4me3 itChIP-seq signals.

-- Why is the insulation score both positive and negative

Reply: Because the insulation scores are row-scale normalized when clustering in heatmaps.

-- how was it calculated (the methods only refer the reader to another paper...)?

Reply: TAD insulation scores (or called TAD separation scores) were generated when identifying TAD boundaries with hicFindTADs tool of HiCEXplorer software. According to hicFindTADs tool of HiCEXplorer software, TAD insulation score is used to identify the degree of separation between the left and right regions of each Hi-C matrix bin. The insulation score is measured using the z-score of the Hi-C matrix and defined as the mean z-score of all the matrix contacts between the left and right regions.

Figure 1g: I am confused about how a single p-value is reported for all of the comparisons between gene-expression between the violin plots.

Reply: Sorry for this confusion. The single P-value was derived from the Kruskal–Wallis test made for gene expressions across all four stages. In this revision, for more meaningful comparison, we revise it into pairwise comparisons between any two populations through two-sample Wilcoxon Rank Sum test.

Also, the two sentences reporting the p-values seem incomplete: e.g. “Genes in the increasing-strength TAD boundaries, Kruskal-Wallis, $p = 0.0011$ ” is lacking a verb to indicate where the reader attention should be focused. Can the authors please clarify this?

Reply: We now change it to two-sample Wilcoxon Rank Sum test between any two populations as indicated in Figure 1g and legends.

The authors claim that "... H3K27me3... signals were markedly accumulated within TADs". However, the data does not appear to support this claim. In Figure 1d, the profile for H3K27me3 is rather flat within the body of the TAD (i.e. unlike the other profiles, there is no discernible shape). Instead, the data could instead suggest that H3K27me3 is depleted at the TAD boundary more than accumulating within the TAD.

Reply: Overall, we agree with the reviewer that H3K27me3 signals were markedly lower at the TAD boundary and relatively accumulating within the TAD. As such, the main text is also modified in this revision.

-- The last point to note regarding H3K27me3 is that is that the itChIP-seq signals are the least reproducible (e.g. showing only ~60-70% Pearson correlations between replicates, and sometimes higher correlations with other cell types). Due to the poor reproducibility of H3K27me3 signals with itself, I suggest that the authors be more cautious in their statement, or revise it to be more consistent with their data.

Reply: Thanks for suggestions. Supplementary Figure 1d present the genome-wide correlations in H3K27me3 between samples. As the reviewer pointed out, the correlation values range from 63-78%, lower than other histone modifications. However, since they passed our quality control (replicates with >95% overlap peaks in the same sequencing depth), we kept them for further analysis. Nevertheless, we thank the reviewer and are more cautious in detailed conclusions with H3K27me3 in the main text and figure legends.

For Figures 1e-g why did the authors choose the top 1000 variable TADs? How were these variable TADs defined?

Reply: In Figures 1e-g, we chose the top 1000 variable TAD boundaries, based on boundary strengths.

First, TAD boundaries at four cell stages were merged together, giving rise to 4839 TADs in total with domain scores calculated from qualified and validated Hi-C pairs.

In the analysis of variable TADs (shown later), we found that there were 2,068 TADs with domain scores with FDR < 0.001, determined as significantly varied during development. To explore the dynamics of histone modification along with TAD domain score changes, we chose top 1000 TADs ranked by variances of domain scores across four stages from 2,068 TADs.

Based on this, in accordance with top 1000 variable TADs for detailed analysis, we also choose the top 1000 variable TAD boundaries. We calculated the standard deviations of the strength of each TAD boundary across four cell stages and sorted boundaries by standard deviations. The top 1000 variable TAD boundaries were selected based on the ranking order.

-- How would the results change if the authors had chosen the top 2000, or 3000 TADs, or only the top 100 TADs? How robust are these results? Can the same conclusions be drawn if instead all TADs are used as in Supplementary Figure 1g?

Reply: Indeed, the same conclusions can be drawn. We perform analysis with all TAD boundaries and top 200 variable TAD boundaries. The results are shown as following (Reviewer figure 3).

Reviewer figure 3. Clustering of all TAD boundaries or top 200 variable TAD boundaries. a Clustering of all TAD boundaries by strengths, with H3K4me3 signals on the right correspondingly. **b** Clustering of top 200 variable TAD boundaries by strengths.

Supplementary Fig. 1f and Reviewer figure 3a focus on all TAD boundaries. Among clustering of all TAD boundaries, C2 boundaries with decreased strengths are related with basic cell behaviors. C3 boundaries with increased strengths do not yield any significant biological processes.

Reviewer figure 3b is centered on top 200 variable TAD boundaries. We calculated the standard deviations of the strength of each TAD boundary across four cell stages and sorted boundaries by standard deviations. The top 200 variable TAD boundaries were selected based on the ranking order. The cluster with increased boundary strength does

not yield significant biological processes. The cluster with decreased boundary strength is related with transcription activity.

Related to Figure 1e, the main text says: “This indicated a role implicated in HSC dormancy, as reported elsewhere”. What element is playing the role (is the role for H3K4me3, or boundary strength, or something else)?

Reply: Sorry for the confusion. We meant that the TAD boundary strength may play a role implicated in HSC dormancy.

The author’s conclusion remark: “these data support that the TAD boundary dynamics is dramatic, facilitating the metabolism activity for hematopoiesis” is not supported by the data. The authors have only shown a partial correlation between changes in gene expression (e.g. from a seemingly selectively chosen GO-term expression analysis) and changes to boundary strengths. Indeed, most cells do not show any significant sign of change in gene expression – e.g. even in the author’s own data it only occurs between LT versus the other cell types. I strongly suggest the authors revise this statement or provide more direct evidence (some suggestions below).

Reply: Thanks for these important comments. We carefully address them as below.

Suggestions to the point above:

1a) To show what fraction of TAD boundaries undergo significant changes: how much of the change is due to noise in the data versus intrinsic to the biological system? The authors could test this, for instance, by downsampling or re-sampling the data, and comparing across conditions or within conditions.

Reply: These suggestions are well-taken in the revision. Before comparing Hi-C data, we normalize the sequencing depth by down-sampling the data to the smallest matrix. After re-sampling the data, the TAD boundaries identified and the strengths are nearly the same. Also, we choose the top variable TAD boundaries to minimize the potential noise in the data.

1b) The authors used the insulation score to define TAD boundaries, which does not distinguish between compartments and TADs – can the authors differentiate between contributions due to A/B compartments and TAD boundaries? If so, how did they do it?

Reply: The insulation scores are calculated from 40 kb bin Hi-C matrices, while compartments are defined with 100 kb bin Hi-C matrices. When calculating insulation scores, the background in the neighborhood will be considered, no matter if the TAD boundary is within compartment boundaries. Also, no matter whether compartment boundaries and TAD boundaries are overlapped, the Hi-C reads are actually detected and used for calculating the changing insulation ability. The compartments and TAD

boundaries are measured in different scales. There is no need to separate these two objects.

2) *In the GO-term analysis what p-value is even deemed significant when multiple tests are being performed (i.e. what is the Bonferroni – or other - corrected p-value?)*

Reply: P -value < 0.05 after BH correction is considered as significant.

3) *As far as the claim that TAD dynamics “facilitate the metabolism activity for hematopoiesis”, if this statement is to be made, perhaps the authors could look at other cell types in data available elsewhere – is there any evidence in any cell type that TAD insulation is strong in these regions in the absence (or presence) of gene expression?*

Reply: We thank the reviewer for this important point. Indeed, we did not observe significant changes in transcription associated with dynamic TAD boundaries across four populations. Since metabolic pathways have been established to be involved in hematopoiesis across many stages⁵, we reasoned that no significant changes in transcription likely lie with two possibilities: 1. Transcriptional alteration may take place later beyond this developmental window than these boundary strength changes as detected here; 2. We cannot exclude the current technical limitation with scRNA-seq datasets from each population. Therefore, we modified our description to “Despite of no detected alteration in transcription, examination of the TAD boundary strength dynamics may help predict future changes in metabolic pathways required for hematopoiesis”.

We also curate evidence as reported elsewhere in other cell types supporting that TAD insulation is strong in the regions in the presence of some gene expressions.

PMID: 29053968 (Boyan Bonev, et al., Cell. 2017)¹⁸: In the neural differentiation system (mESC, NPC, CN), it supported that transcription was correlated with but not sufficient to cause insulation at TAD boundaries *de novo*.

PMID: 22495300 (Jesse R. Dixon, et al., Nature. 2012)¹⁹: In ESC system, factors associated with active promoters and gene bodies were enriched at boundaries in both mouse and humans. Boundaries of topological domains were enriched for the insulator binding protein CTCF, housekeeping genes, transfer RNAs and SINE retrotransposons. Indeed, boundaries associated with both CTCF and a housekeeping gene account for nearly one-third of all topological boundaries in the genome.

PMID: 26518482 (Sergey V Ulianov, et al., Genome Res. 2016)²⁰: In four cell lines of various origins (S2, Kc167, DmBG3-c2, and OSC), this study showed that active chromatin and transcription played a key role in chromosome partitioning into topologically associating domains.

In Supplementary Figure 2a, are the different columns representing technical or biological replicates? Please specify.

Reply: Biological replicates. We also add the notes in figure legends.

For Figs. 2a-b again, why are only 1000 TADs chosen

Reply: TAD boundaries from four cell stages of samples were merged together and the regions between every two adjacent boundaries were TADs for further analysis. There were 4,839 TADs in total with domain scores calculated from qualified and validated Hi-C pairs. We first applied ANOVA analysis with Benjamini-Hochberg correction to domain scores of every TAD across four cell stages to judge the variation of every TAD. We found that there were 2,068 TADs with domain score with the cutoff $FDR < 0.001$, determined as significantly varied during development. To explore the dynamics of histone modification along with TAD domain score changes, we chose top 1000 TADs ranked by variances of domain scores across four stages from 2,068 TADs.

-- how do the results change with other numbers of TADs?

Reply: As we show above, our overall conclusions do not change. We re-analyze all TADs (Supplementary Fig. 2b) and the 2,068 TADs with domain scores with the cutoff $FDR < 0.001$ (Supplementary Fig. 2c-f) in this revision. Among the 2,068 significantly changing TADs, those with increased domain scores are still related with immune and hematopoiesis processes as with top 1000 variable TADs. In the meantime, those with decreased domain scores at a high level are related with multiple non-hematopoiesis tissue developments. This is also the case with top 1000 variable TADs.

Supplementary Fig. 2b Hierarchical clustering of all TADs by normalized domain scores, alongside histone modifications in ChIP-seq peaks within TADs.

Supplementary Fig. 2c-f

Supplementary Fig. 2c-f: **c** Hierarchical clustering of the 2,068 TADs with the cutoff $FDR < 0.001$ by normalized domain scores. The clustering was performed with row-z-scaled values. **d** Further Hierarchical clustering of 1,020 TADs with decreased domain scores by normalized domain scores (no row scaled). **e** Biological processes in GO analysis of 515 TADs with decreased but high domain scores. **f** Biological processes in GO analysis of 982 TADs with increased domain scores.

-- What are the corrected p -values for multiple hypothesis testing significance for the GO-term analysis?

Reply: Thanks for this point. I re-calculate the corrected P -values (BH correction) and revise the information in the figures (Fig. 2d).

Fig. 2d

Fig. 2d Biological processes enriched in three main clusters of TADs. The GO analysis was done by GREAT. *P*-values were corrected by BH correction.

The statement “C1 with decreased TAD connectivity was accompanied with H3K27ac increase and H2K27me3 reduction” is seemingly inconsistent with the figure 2 a-b. From the pre to LT columns, H3K27ac goes from more red to more blue suggesting a decrease. Did the authors mean H3K27ac decrease? Does that change the author’s conclusions?

Reply: Thanks for pointing out this typo. It should be “C1 with decreased TAD connectivity was accompanied with H3K27ac decrease”.

Generally, the comparisons in the paragraph: “C2 with a sharp decrease in TAD connectivity between early AECs and HECs displayed higher H3K27me3 and lower H3K4me1 and nearly complete depletion of H3K27ac and H3K4me3. C3 with a gradually increasing TAD connectivity showed strong H3K27ac, H3K4me1 and H3K4me3 signals as early as eAECs with a slow reduction in H3K27me3. Interestingly, genes within C3 were associated with several hematopoiesis related functional terms...” needs quantification.

Reply: Thanks for this important suggestion. The quantifications of the domain score and histone modifications in three clusters are added in Fig.2c (mean values) and Reviewer figure 4.

Fig. 2c

Fig. 2c Quantification curves for the asynchronous change of feature histone modifications within TADs and TAD domain scores in three main clusters. Mean values were calculated for TADs in each cluster.

a

ANOVA test within each cluster

	H3K27ac	H3K27me3	H3K4me1	H3K4me3
C1	0.000875 ***	0.982	0.646	0.153
C2	0.17	0.0416 *	7.36e-08 ***	0.0081 **
C3	4.14e-06 ***	4.36e-14 ***	3.36e-08 ***	0.265

Reviewer figure 4. Statistical quantification of TAD clusters as in Fig. 2.

a The ANOVA P -values among four stages of four histone modification signals in every cluster were shown.
b Violin plots showing changes of domain scores and histone modifications in three cluster TADs. The $\log_2(\text{fold change})$ values of significantly changing histone modifications were calculated between two adjacent stages.

-- How significant are the differences that the authors highlight? More importantly, what are the fold-changes in the signal values?

Reply: As mentioned above, we add this quantification in Figure 2c in this revision.

The conclusions were described as following. C1 with decreased TAD connectivity was accompanied with H3K27ac decrease obviously from pre-HSCs to LT-HSCs. C2 with a sharp decrease in TAD connectivity between eAECs and HECs displayed increasing H3K27me3 and H3K4me1 (low level), with nearly complete depletion of H3K27ac and H3K4me3. C3 with a gradually increasing TAD connectivity showed strong H3K27ac, H3K4me1 and H3K4me3 signals as early as eAECs (Fig. 2b and Supplementary Fig. 2c). Specifically, H3K27ac significantly increased during the pre-HSC to LT-HSC transition, companied by a gradual reduction in H3K27me3.

“Specific inspection of TAD scores and histone modifications at the Runx1 exemplified the increased intra-TAD contacts. This identified the newly engaged P1 promoter in HECs evidenced by H3K4me3 signals and the enhancer activation by H3K27ac signals

(Fig.2e).” It appears that much of what the authors are showing in the insets (i.e. higher Hi-C signal) can be attributed to the change in the contact probability decay genome-wide that occurs between eAECs and the later cell types (see Fig. 2f). Can the authors compute the observed over expected Hi-C scores and show that specific interactions appear after removing the contribution due to the changing overall contact probability scaling with genomic distance?

Reply: Thanks for the suggestions, we compute the observed over expected Hi-C matrices to remove the contribution due to the changing overall contact probability scaling with genomic distance. The following figure exemplifies specific interactions appearing from eAECs to LT-HSCs, with observed over expected matrices (Fig. 2g,h and Supplementary Fig. 3a).

Fig. 2g

Fig. 2g Exemplification showing the increasing intra-TAD interactions with observed/expected Hi-C matrices (5 kb resolution) at *Runx1* gene site, associated with feature histone modifications.

Fig. 2h

Fig. 2h Curves quantifying the changing loop strength between the *Runx1* P1 promoter and neighboring regions. The genomic coordinate is chr16:92,614,120-93,400,000, from -212 kb to +574 kb relative to *Runx1* P1. The observed/expected Hi-C matrices at 5 kb resolution were used for calculation. The *Runx1* P1 region in red was TSS \pm 2.5 kb. The region 1 in yellow overlapped with *Runx1* P2 promoter, while the region 2 in green represented a distal enhancer.

Supplementary Fig. 3

Supplementary Fig. 3: Examples of changing intra-TAD connectivity and multiple histone modifications.
a Contact heatmap exemplifying short-range contacts and long-range contacts around *Runx1* gene within TADs, enhanced from eAEC through HEC and pre-HSC to LT-HSC. The boxes indicate examples of specific interactions increasing or appearing from eAECs to LT-HSCs. Visualization was performed with observed/expected Hi-C matrices at 5 kb resolution from a large view. **b** Exemplification showing the increasing intra-TAD interactions associated with feature histone modifications at *Runx1* gene site. This visualization was performed with depth-normalized Hi-C matrices (5 kb resolution) with observed treatment. **c** Quantification curves for the changing loop strength between the *Runx1* P1 promoter and neighboring regions. The quantification was performed from normalized Hi-C matrices (5 kb resolution) with observed signals. The *Runx1* P1 region in red was TSS \pm 2.5 kb. The region 1 in yellow overlapped with *Runx1* P2 promoter, while the region 2 in green represented a distal enhancer. **d** Bar plot showing the relative interaction strengths between *Runx1* P1 and two regions as indicated in yellow or green in (c). eAEC, early AEC; pre, pre-HSC; LT, LT-HSC.

-- Perhaps deeper sample sequencing may be required to reveal a specific looping-type interaction. Can the authors quantify (similarly to Supl. Fig. 3c) the change in the loop strength at *Runx1* – what is the effect size/ fold-change over time?

Reply: Thanks for this comment. We apply the same analysis as in Supplementary. Fig. 3c, to quantify the changing loop strength between the *Runx1* P1 promoter and neighboring regions at 5 kb resolution. Further, the interaction strength between the *Runx1* P1 and two regions as indicated is quantified (Supplementary Fig. 3b-d).

Figure 3a – how did authors identify loops – were all the data pooled together?

Reply: The replicates of Hi-C matrices at the 10 kb resolution were normalized to the sequencing depth of the smallest replicate. All replicates of each cell type were pooled. The sequencing depths of four cell populations were further normalized to the smallest one. Next, the pooled and normalized Hi-C matrices at the 10 kb resolution were used for loop calling.

The authors used a loop calling algorithm, but loop calling is generally very difficult – especially with noisy data with only a few million reads. Can the authors show specific examples of good and bad loops called by their data? What fraction of loops called by their algorithm usage were nicely visible in their Hi-C maps?

Reply: Thanks for these suggestions. In our Hi-C data, the total read pairs are 280-700 M for each replicate. After mapping, filtering and de-duplication, the non-duplicated read pairs used for pooled or separately analysis range from 50 to 170 M for each replicate (shown in the following table). More details about the data depth of Hi-C are shown in the Supplementary Table 1. When we call loops with hicDetectLoops in HiCExplorer, we use the pooled matrices merged by depth-normalized replicates of each cell population. The pooled matrices are further normalized for depth among different cell populations. Finally, the validated, non-duplicated and normalized reads used for loop calling are around 250 M. As for those with more final reads up to 750 M, we compare loops called from all 750 M or the normalized 250 M. The loops with good quality are mostly in common. Our final validated and non-duplicated read pairs are comparable and even more than previous reports, such as 20-150 M non-duplicated read pairs in PMID: 28709003 (Yuwen Ke, et al., Cell, 2017)²¹, 20-80 M non-duplicated read pairs in PMID: 29466755 (Hu et al., Immunity, 2018)¹⁷, and 20-50 M just unique read pairs in PMID: 31253982 (Irene Miguel-Escalada, et al., Nature Genetics, 2019)²².

Part of Supplementary Table 1. Data quality control of Hi-C libraries.

Stage	Replicate	Final reps	Cell number	Nonduplicated Read pairs	Stage	Replicate	Final reps	Cell number	Nonduplicate d Read pairs
AEC	AEC-hic-rep1_part1	AEC-rep1	1000	65.1 M	pre-HSC	reHSC-hic-rep	preHSC-rep1	1068	168.7 M
	AEC-hic-rep1_part2		1000	58.9 M		reHSC-hic-rep	preHSC-rep2	1083	137.4 M
	AEC-hic-rep2_part1	AEC-rep2	1000	60.0 M		reHSC-hic-rep	preHSC-rep3	1100	154.6 M
	AEC-hic-rep2_part2		1000	63.7 M		reHSC-hic-rep	preHSC-rep4	1368	122.5 M
	AEC-hic-rep3_part1	AEC-rep3	1000	32.6 M		reHSC-hic-rep	preHSC-rep5	963	169.3 M
	AEC-hic-rep3_part2		1000	122.8 M					
HEC	HEC-hic-rep1_part1	HEC-rep1	266	48.17 M	LT-HSC	.THSC-hic-rep	LTHSC-rep1	576	99.0 M
	HEC-hic-rep1_part2		539	72.30 M		.THSC-hic-rep	LTHSC-rep2	636	90.0 M
	HEC-hic-rep2	HEC-rep2				.THSC-hic-rep	LTHSC-rep3	637	93.3 M
			911	125.5 M		.THSC-hic-rep	LTHSC-rep4	832	92.1 M

We validate our loops by aggregation analysis. The quantification results of good loops are shown in Supplementary Fig. 4a,b. They are examples of good loops for further analysis, with peak patterns in the centers and cell-type specificity. Our four cell populations are similar, so the cell-type specificity reflects on the differential strengths. If

loops are bad, they would not display evident peak patterns in the center (Supplementary Fig. 4b).

Supplementary Fig. 4a

Supplementary Fig. 4b

Supplementary Fig. 4a Heatmaps showing the aggregate peak analysis (APA) signals on loops identified in each cell population. The Hi-C matrices at 10 kb resolution were used for loop calling.
Supplementary Fig. 4b Curves quantifying the APA signals on loops identified in each cell population, with Hi-C signals of four cell populations. The Hi-C matrices at 10 kb resolution were used for loop calling.

It is hard to judge the visible quality of loops one by one to derive a specific fraction. Instead, we estimated the fraction of good loops by adjusting the parameter “pit” during loop calling. “pit” means the minimum number of interactions, with which a detected peak will be considered. If we set “pit” as 5 or a higher value, the numbers of loops are 5,000-7,000. If we set “pit” as 3, the numbers are 6,000-10,000. If we set “pit” as 1, the numbers are 26,000-28,000. We use loops called by “pit 10”, with reliable interaction strengths.

In the figure 3a-3b colour maps, what are the colour bar units, and how were the values calculated/what do they represent, e.g. $\log_2(\text{value})$, $\log_{10}(\text{value})$, $\text{value}/\text{input}$?

Reply: In Fig.3a, the color bar unit is the loop strength (calculated by normalized CPM) normalized by row scale. In Fig.3b, the color bar units of histone modifications are normalized ChIP-seq signals (reads) on the corresponding loop anchors. Histone modification ChIP-seq reads are normalized to the same level of sequencing depth and used for calculating signal matrices of the heatmap. In Fig.3b, the color bar unit of gene expression heatmap is the expression value expressed as $\log_2(\text{TPM}/10+1)$ and normalized by row scale.

In Figure 3e, how are the “interactions” on top of the tracks defined? The authors refer to a software package in the methods that generate them, but what do the results mean? Are the shown interactions (or changes to interactions) robust? Was Hi-C used to generate the arches (shown on top of the tracks), or was the track view generated only from the itChIP-seq tracks directly?

Reply: The interactions are identified with the hicDetectLoops function in HiCEXplorer. The robustness of the interactions by Hi-C were validated by multiple tests with the same results from varying parameters. Indeed, both Hi-C and itChIP-seq data were used for generating these tracks.

In Figure 3f, why is the violin plot showing negative values of gene expression?

Reply: As we mentioned above, we fully examined the calculation and found that the “negative values” shown in figures were due to the confusing graphic presentations generated by ggpubr package, with ‘trim’ function set as ‘FALSE’.

First, we checked the $\log_2(\text{TPM}/10+1)$, confirming that there are no negative values. The matrices of $\log_2(\text{TPM}/10+1)$ are also attached. Second, we looked back to the source codes of the ggpubr package. The ggpubr package utilizes the ggplot package. The parameter “trim” is TRUE by default in the ggplot package. However, the parameter “trim” is FALSE by default in the ggpubr package. If TRUE, “trim” function will trim the tails of the violins to the range of the data. If FALSE, it does not trim the tails. The tails are generated by smoothing during fitting the distribution. For clarification, we set the parameter “trim” to TRUE (Fig. 3f).

Fig. 3f Violin plots showing expression of representative genes in three clusters with $\log_2(\text{TPM}/10+1)$. Wilcoxon Rank Sum test was performed between two adjacent stages. *, P -value <0.05; **, P -value <0.01; ***, P -value <0.001; ****, P -value <0.0001; “ns”, not significant.

-- The authors establish a “chromatin looping hub” for RUNX1 using Cytoscape. Can the authors explain how this graphical network helps establish that “core chromatin interactions among some hematopoiesis genes initiated at early AECs become stronger were likely mediated by steadily increasing RUNX1 expression and binding signals at later developmental stages”?

Reply: From the visualized RUNX1-engaged E-P interactions in Fig.4f, we revise and tone down the conclusions as follows. “Taken together, our results indicated RUNX1-engaged E-P interactions promoted hematopoiesis early as in eAECs, which were reinforced within EHT. Presumably, RUNX1-engaged E-P interactions might participate in suppressing non-hematopoiesis gene programs, but not further explored in this study.”

Two main clusters of genes in RUNX1-engaged E-P interactions are C2 (hematopoiesis processes) and C4 (non-hematopoiesis processes). From Fig4c-e, C2 shows good correlation in gene expression, RUNX1 occupancy and histone modifications, promoting hematopoiesis early in eAECs. The interactions among C2 genes are found in a larger number and stronger degrees. From Fig.4c-e, C4 shows expression decrease in non-hematopoiesis processes, accompanied by more complexed change trends in RUNX1 occupancy and histone modifications.

-- The analysis seems tautological – the authors first identify a hub via RUNX1, then label interactions within the hub, then look at changing interactions within a hub, seeing that some become stronger within a specific subset within the hub, and use this to claim that the increasing strength of interaction is “likely mediated by steadily increasing RUNX1 expression”. I strongly suggest this analysis to be removed and the conclusions revisited, unless it can be appropriately explained, independently verified and quantitatively supported.

Reply: Thanks for this specific suggestion. Instead, we add sub-clustering and quantification analysis to explore how RUNX1-engaged E-P interactions among C2 genes change in Fig. 4. The new analysis in Fig. 4f,g shows that RUNX1-engaged E-P interaction strengths were enhanced within the gene loci linked to definitive hematopoiesis while decreased within other loci related to mitotic cell cycle and angiogenesis during development.

“Sub-clustering by the interaction strengths showed that RUNX1-engaged E-P interactions with decreased strengths were linked to mitotic cell cycle and angiogenesis (Fig. 4f and Supplementary Fig. 6a). The interaction strengths within the gene loci enriched for the GO terms, B cell proliferation and interferon α/β signaling, appeared transiently higher in HECs and pre-HSCs (Fig. 4f and Supplementary Fig. 6a). Interestingly, the increasing sub-cluster was related to T cell activation, adaptive immune and definitive hemopoiesis (Fig. 4f and Supplementary Fig. 6a). Statistical analyses were further performed in Fig. 4g.”

Fig. 4f

Fig. 4g

Fig. 4f Lines reflecting strengths of three sub-clusters of RUNX1-engaged E-P interactions among C2 genes. Those RUNX1-engaged E-P interactions existing from eAEC to LT-HSC were used for clustering. The black lines were fitted for the interaction dynamics in each cluster. The thick white bars represented a 95% confidence interval.

Fig. 4g Violin plots quantifying the strengths of three sub-clusters of RUNX1-engaged E-P interactions among C2 genes, corresponding to (f). Box-and-whiskers plots represented the maxima, 75th percentile, median, 25th percentile, and minima. The two-sample Wilcoxon Rank Sum test was performed between two adjacent stages. *, *P*-value <0.05; **, *P*-value <0.01; ***, *P*-value <0.001; ****, *P*-value <0.0001; “ns”, not significant.

Supplementary Fig. 6a

Supplementary Fig. 6a Biological processes in Gene Ontology analysis with genes in three sub-clusters of C2 RUNX1-engaged E-P interactions in Fig.4f.

Why define a looping hub only with C2 genes when RUNX1/H3K27ac signals are also well correlated in the other clusters?

Reply: Because most C2 genes are related to hematopoiesis. Moreover, not only RUNX1/H3K27ac signals but also gene expressions are well correlated in C2. In this revision, we only analyze C4 in Supplementary Fig. 4.

In Figure 5a, how is “distal” defined? What is the genomic distance?

Reply: With genomic distance to TSS > 5 kb is defined as “distal”.

In Fig. 5, the authors use the term “RUNX1-mediated interactions”. How can the authors make this claim? What is the evidence supporting that RUNX1 is casual (i.e. mediating) the interactions?

Reply: Thanks for this important correction. For clarity, “RUNX1-mediated interactions” should change to “RUNX1-engaged interactions”.

1. Zhu Q, *et al.* Developmental trajectory of prehematopoietic stem cell formation from endothelium. *Blood* **136**, 845–856 (2020).
2. Fadlullah MZ, *et al.* Murine AGM single-cell profiling identifies a continuum of hemogenic endothelium differentiation marked by ACE. *Blood*, (2021).
3. Beyer S, Kelly RG, Miquerol L. Inducible Cx40-Cre expression in the cardiac conduction system and arterial endothelial cells. *Genesis* **49**, 83–91 (2011).
4. Hou S, *et al.* Embryonic endothelial evolution towards first hematopoietic stem cells revealed by single-cell transcriptomic and functional analyses. *Cell Res* **30**, 376–392 (2020).
5. Nakamura-Ishizu A, Ito K, Suda T. Hematopoietic Stem Cell Metabolism during Development and Aging. *Dev Cell* **54**, 239–255 (2020).
6. Iizuka K, *et al.* Lack of Phenotypical and Morphological Evidences of Endothelial to Hematopoietic Transition in the Murine Embryonic Head during Hematopoietic Stem Cell Emergence. *PLoS One* **11**, e0156427 (2016).
7. Tober J, Yzaguirre AD, Piwarzyk E, Speck NA. Distinct temporal requirements for Runx1 in hematopoietic progenitors and stem cells. *Development* **140**, 3765–3776 (2013).
8. Chen MJ, Yokomizo T, Zeigler BM, Dzierzak E, Speck NA. Runx1 is required for the endothelial to haematopoietic cell transition but not thereafter. *Nature* **457**, 887–891 (2009).
9. North TE, *et al.* Runx1 expression marks long-term repopulating hematopoietic stem cells in the midgestation mouse embryo. *Immunity* **16**, 661–672 (2002).
10. Okuda T, van Deursen J, Hiebert SW, Grosveld G, Downing JR. AML1, the target of multiple chromosomal translocations in human leukemia, is

- essential for normal fetal liver hematopoiesis. *Cell* **84**, 321-330 (1996).
11. Yzaguirre AD, de Bruijn MF, Speck NA. The Role of Runx1 in Embryonic Blood Cell Formation. *Adv Exp Med Biol* **962**, 47-64 (2017).
 12. Sandler VM, *et al.* Reprogramming human endothelial cells to haematopoietic cells requires vascular induction. *Nature* **511**, 312-318 (2014).
 13. Wilson NK, *et al.* Combinatorial transcriptional control in blood stem/progenitor cells: genome-wide analysis of ten major transcriptional regulators. *Cell Stem Cell* **7**, 532-544 (2010).
 14. Feld C, *et al.* Combined cistrome and transcriptome analysis of SKI in AML cells identifies SKI as a co-repressor for RUNX1. *Nucleic Acids Res* **46**, 3412-3428 (2018).
 15. Gilmour J, Assi SA, Noailles L, Lichtinger M, Obier N, Bonifer C. The Co-operation of RUNX1 with LDB1, CDK9 and BRD4 Drives Transcription Factor Complex Relocation During Haematopoietic Specification. *Sci Rep* **8**, 10410 (2018).
 16. Chandra T, *et al.* Global reorganization of the nuclear landscape in senescent cells. *Cell Rep* **10**, 471-483 (2015).
 17. Hu G, *et al.* Transformation of Accessible Chromatin and 3D Nucleome Underlies Lineage Commitment of Early T Cells. *Immunity* **48**, 227-242. e228 (2018).
 18. Bonev B, *et al.* Multiscale 3D Genome Rewiring during Mouse Neural Development. *Cell* **171**, 557-572. e524 (2017).
 19. Dixon JR, *et al.* Topological domains in mammalian genomes identified by analysis of chromatin interactions. *Nature* **485**, 376-380 (2012).
 20. Ulianov SV, *et al.* Active chromatin and transcription play a key role in chromosome partitioning into topologically associating domains. *Genome Res* **26**, 70-84 (2016).
 21. Ke Y, *et al.* 3D Chromatin Structures of Mature Gametes and Structural Reprogramming during Mammalian Embryogenesis. *Cell* **170**, 367-381. e320 (2017).
 22. Miguel-Escalada I, *et al.* Human pancreatic islet three-dimensional chromatin architecture provides insights into the genetics of type 2

diabetes. *Nat Genet* **51**, 1137–1148 (2019).

VIEWERS' COMMENTS

Reviewer #1 (Remarks to the Author):

The authors have addressed all minor points and most of the main points satisfactorily. However, I still have queries regarding main points 1 and 3 that should be addressed.

Main point 1 - The nature of the earlyAEC. The authors cite the recent papers by Zhu et al., *Blood* 2020 (Speck and Tan groups) and Fadlullah et al., *Blood* 2021 (Lacaud group) as support that a transition from AEC to HE still occurs in the E10.5 dorsal aorta. However, both papers mainly focus on the transitions of pre-HE to HE and beyond. I do not think that the presence of AE in those single cell datasets/trajectories can be interpreted as indicating that the AE to pre-HE transition still occurs at this timepoint. As the authors agree, this will require future lineage tracing. While the functional assays performed on Lin⁻ AGM populations with or without ACE+Cdh5⁺ cells (Fadlullah et al., 2021) do show that hematopoietic potential resides in the Ace+Cdh5⁺ population, the assays used do not specifically test for hemogenic potential of AECs. Indeed, the ACE⁺ cells are a mix of EC, pre-HE, HE, EHT cells and IAHC (Fig 4G and Suppl. Fig 15F), precluding conclusions specifically about the hemogenic potential of the E10.5 AEC. Although I am not familiar with the cultures used (hanging drop and coculture with endothelial cells generated in the Rafii group) these do not appear to have been developed to specifically probe hemogenic potential of endothelium. For this the OP9 co-cultures are still the most widely used assay (see e.g. Eilken et al *Nature* 2009). Similarly, the Cx40-Cre line cited by the authors cannot conclusively show that AECs still transition to HECs at E10, as Cx40 (Gja5) is also expressed by HEC as reported by Zhu et al., *Blood* 2020 (VisCello app), and the authors' own work in Hou et al., *Cell Research* 2020, Fig1C. In the latter paper the authors nicely identified eAEC and lateAEC from E9.5 to E11 embryos, and in Figure 1i show that while all E9.5 eAEC map to the HEC specification branch of the pseudotime trajectory plot, only a part of the E10 and none of the E10.5 or E11 eAECs do. As in the current manuscript embryos between 31 and 35sp were used (i.e. E10 to E10.5), I think it is plausible that at least part of the eAEC analysed in the present paper will not become HECs. While the authors have a point that for consistency with their previous work they should continue to call their cells eAEC, I remain of the opinion that they should qualify their conclusions and acknowledge that clear differences observed between E10 AECs and the E10/E11 EHT populations may not solely reflect an important transition in EHT, but can also be due to at least part of the E10 AEC being on a different fate trajectory from the E10 HE/pre-HSCs. Acknowledging this will not diminish the relevance and interest of the paper.

Main point 3. This point relates to main point 1, in querying the nature/heterogeneity of the eAEC cell population.

The authors nicely demonstrate the biological relevance of their Runx1 ChIP-seq in the revision, satisfactorily showing that the Runx1 binding is not due to background. This leaves the question whether the Runx1 binding in the eAEC could be due to this population being a mix of HE and eAEC. To my knowledge, there is no data in the literature to suggest that non-HE expresses Runx1. The authors appear to suggest in the rebuttal that Runx1 expression in the dorsal side of the aorta may be in non-HE. I am not sure what the functional evidence for this is. The Medvinsky lab showed that the dorsal side of the aorta contains CFU-C activity (Taoudi and Medvinsky, *PNAS* 2007). Endothelial Runx1 expression in that location fits with this, as Runx1 is not specific for EHT in the HSC lineage and is also required for EHT into hematopoietic progenitors (e.g. Tober et al., *Development* 2013). The possibility of the the eAEC being a heterogeneous population in terms of cell type/fate should be acknowledged.

Reviewer #2 (Remarks to the Author):

The reviewer's concerns had been addressed. I have no more comments and recommend it to be published.

REVIEWERS' COMMENTS

Reviewer #1 (Remarks to the Author):

The authors have addressed all minor points and most of the main points satisfactorily. However, I still have queries regarding main points 1 and 3 that should be addressed.

Main point 1 - The nature of the earlyAEC. The authors cite the recent papers by Zhu et al., Blood 2020 (Speck and Tan groups) and Fadlullah et al., Blood 2021 (Lacaud group) as support that a transition from AEC to HE still occurs in the E10.5 dorsal aorta. However, both papers mainly focus on the transitions of pre-HE to HE and beyond. I do not think that the presence of AE in those single cell datasets/trajectories can be interpreted as indicating that the AE to pre-HE transition still occurs at this timepoint. As the authors agree, this will require future lineage tracing. While the functional assays performed on Lin- AGM populations with or without ACE+Cdh5+ cells (Fadlullah et al., 2021) do show that hematopoietic potential resides in the Ace+Cdh5+ population, the assays used do not specifically test for hemogenic potential of AECs. Indeed, the ACE+ cells are a mix of EC, pre-HE, HE, EHT cells and IAHC (Fig 4G and Suppl. Fig 15F), precluding conclusions specifically about the hemogenic potential of the E10.5 AEC. Although I am not familiar with the cultures used (hanging drop and coculture with endothelial cells generated in the Rafii group) these do not appear to have been developed to specifically probe hemogenic potential of endothelium. For this the OP9 co-cultures are still the most widely used assay (see e.g. Eilken et al Nature 2009). Similarly, the Cx40-Cre line cited by the authors cannot conclusively show that AECs still transition to HECs at E10, as Cx40 (Gja5) is also expressed by HEC as reported by Zhu et al., Blood 2020 (VisCello app), and the authors' own work in Hou et al., Cell Research 2020, Fig1C. In the latter paper the authors nicely identified eAEC and lateAEC from E9.5 to E11 embryos, and in Figure 1i show that while all E9.5 eAEC map to the HEC specification branch of the pseudotime trajectory plot, only a part of the E10 and none of the E10.5 or E11 eAECs do. As in the current manuscript embryos between 31 and 35sp were used (i.e. E10 to E10.5), I think it is plausible that at least part of the eAEC analysed in the present paper will not become HECs. While the authors have a point that for consistency with their previous work they should continue to call their cells eAEC, I remain of the opinion that they should qualify their conclusions and acknowledge that clear differences observed between E10 AECs and the E10/E11 EHT populations may not solely reflect an important transition in EHT, but can also be due to at least part of the E10 AEC being on a different fate trajectory from the E10 HE/pre-HSCs. Acknowledging this will not diminish the relevance and interest of the paper.

Reply: We are grateful for these constructive comments and suggestions to make the interpretations in this work more accurate. We also fully concur with the reviewer for these alternative interpretations for the heterogeneity nature of eAECs as assayed and defined in this manuscript. Specifically, we acknowledge this statement in the discussion that at least part of the eAECs analyzed in the present paper will not become HECs, instead may

take on a different fate trajectory, though the difference between eAECs and HECs was examined for the EHT process in the current work.

Main point 3. This point relates to main point 1, in querying the nature/heterogeneity of the eAEC cell population.

The authors nicely demonstrate the biological relevance of their Runx1 ChIP-seq in the revision, satisfactorily showing that the Runx1 binding is not due to background. This leaves the question whether the Runx1 binding in the eAEC could be due to this population being a mix of HE and eAEC. To my knowledge, there is no data in the literature to suggest that non-HE expresses Runx1. The authors appear to suggest in the rebuttal that Runx1 expression in the dorsal side of the aorta may be in non-HE. I am not sure what the functional evidence for this is. The Medvinsky lab showed that the dorsal side of the aorta contains CFU-C activity (Taoudi and Medvinsky, PNAS 2007). Endothelial Runx1 expression in that location fits with this, as Runx1 is not specific for EHT in the HSC lineage and is also required for EHT into hematopoietic progenitors (e.g. Tober et al., Development 2013). The possibility of the eAEC being a heterogeneous population in terms of cell type/fate should be acknowledged.

Reply: This suggestion is well taken in this revision. We agree with the possibility of the eAEC being a heterogeneous population in terms of cell fate potential, which may be due to the heterogeneity of *Runx1* expression, as well as being a mix of HEs and eAECs. This is mentioned in the discussion in this revision.

Reviewer #2 (Remarks to the Author):

The reviewer's concerns had been addressed. I have no more comments and recommend it to be published.

We appreciate this reviewer for his/her previous constructive comments/concerns as well as recognition of the improvement in the last revision.